# Transformers as Implicit State Estimators: In-Context Learning in Dynamical Systems

**Usman Akram** *usman.akram@austin.utexas.edu*
*Department of Electrical and Computer Engineering*
*The University of Texas at Austin*

**Haris Vikalo** *hvikalo@ece.utexas.edu*
*Department of Electrical and Computer Engineering*
*The University of Texas at Austin*

**Reviewed on OpenReview:** *https://openreview.net/forum?id=hIMK5MvGkP*

## Abstract

Predicting the behavior of a dynamical system from noisy observations of its past outputs is a classical problem encountered across engineering and science. For linear systems with Gaussian inputs, the Kalman filter – the best linear minimum mean-square error estimator of the state trajectory – is optimal in the Bayesian sense. For nonlinear systems, Bayesian filtering is typically approached using suboptimal heuristics such as the Extended Kalman Filter (EKF), or numerical methods such as particle filtering (PF). In this work, we show that transformers, employed in an in-context learning (ICL) setting, can implicitly infer hidden states in order to predict the outputs of a wide family of dynamical systems, without test-time gradient updates or explicit knowledge of the system model. Specifically, when provided with a short context of past input–output pairs and, optionally, system parameters, a frozen transformer accurately predicts the current output. In linear-Gaussian regimes, its predictions closely match those of the Kalman filter; in nonlinear regimes, its performance approaches that of EKF and PF. Moreover, prediction accuracy degrades gracefully when key parameters, such as the state-transition matrix, are withheld from the context, demonstrating robustness and implicit parameter inference. These findings suggest that transformer in-context learning provides a flexible, non-parametric alternative for output prediction in dynamical systems, grounded in implicit latent-state estimation.

## 1 Introduction

In-context learning (ICL), particularly in the form of few-shot prompting (Yogatama et al., 2019), has emerged as a prominent research direction in natural language processing (NLP). In this framework, a large language model (LLM) performs new tasks by conditioning on a small number of input–output examples provided in the prompt. One of the earliest works to demonstrate this capability was by Brown et al. (2020), who evaluated GPT-3 across a range of NLP datasets under "zero-shot", "one-shot" and "few-shot" settings. Zhao et al. (2021) identified majority-label bias, recency bias, and common-token bias as sources of instability in GPT-3's accuracy under few-shot prompting, and proposed contextual calibration as a remedy. A theoretical explanation for ICL as a form of implicit Bayesian inference is offered in Xie et al. (2021). Min et al. (2022) explore why ICL works in practice, showing that its success is not dependent on having access to ground-truth labels; it further emphasized the importance of label space, input distribution, and prompt format. In parallel, Schlag et al. (2021) provide a theoretical analysis showing that transformers can act as fast weight programmers.

Early work on using standard transformer decoders for in-context learning of autoregressive models includes (Garg et al., 2022), where the ability of transformers to learn function classes from examples is empirically

investigated. A model is said to learn a function class $\mathcal{F}$ over a domain $\mathcal{X}$ if, for any $f \in \mathcal{F}$ and any inputs $x_1, x_2, \ldots, x_N, x_{\text{query}}$ drawn i.i.d. from $\mathcal{X}$, the model can predict $f(x_{\text{query}})$ given the sequence $x_1, f(x_1), x_2, f(x_2), \ldots, x_N, f(x_N), x_{\text{query}}$. The function classes studied by Garg et al. (2022) range from simple linear functions to sparse linear models, two-layer neural networks, and decision trees.

Two parallel works by Von Oswald et al. (2023) and Akyürek et al. (2023) examine which algorithms transformers implicitly implement when learning function classes in-context. Von Oswald et al. (2023) build on Schlag et al. (2021) and show that linear self-attention layers can be interpreted as performing a step of gradient descent. Specifically, they prove that for a single-headed linear attention layer, there exist key, query, and value matrices such that a forward pass corresponds to one step of gradient descent with an $\ell_2$ loss applied to each token. In contrast, Akyürek et al. (2023) introduce a raw operator framework that supports operations such as matrix multiplication, scalar division, and memory-like read/write. They demonstrate that a single transformer head equipped with suitable key, query, and value matrices can approximate this operator, implying that transformers are, in principle, capable of performing linear regression via either stochastic gradient descent or in closed-form.

Our work investigates whether transformers, using in-context learning (ICL), can learn to perform filtering in dynamical systems described by noisy state-space models. Specifically, we study whether a transformer, conditioned on a short context of past input–output pairs (and, optionally, system parameters), can predict the current output of the system. The model is pre-trained on synthetic trajectories generated by randomly sampled system parameters and evaluated without gradient-based updates at test time. Our goal is to understand which classical filtering algorithms the transformer's behavior most closely resembles under this setup. To that end, we begin by examining the structure of the Kalman filter (Kalman, 1960), the optimal linear estimator for systems with Gaussian noise, and ask whether its operations can be replicated by a transformer architecture. We show that Kalman filtering can indeed be expressed using operations that are readily implemented by a transformer, and provide both a proof-by-construction and empirical evidence that, when sufficiently scaled and provided with enough context, the transformer learns to emulate Kalman-like behavior. Notably, its performance remains strong even when some parameters, such as the state-transition matrix, are omitted from the context, suggesting robustness and an ability to infer missing information implicitly. We then turn our attention to nonlinear systems and empirically demonstrate that transformers can in-context learn to achieve output prediction accuracy comparable to that of classical nonlinear filters such as the Extended Kalman Filter and particle filters. While our analysis and experiments primarily focus on output prediction, we also include preliminary evidence that transformers can, in some cases, explicitly recover latent states, thus highlighting their potential for more general inference tasks.

**Scope and framing.** This paper focuses on the empirical investigation of whether transformers trained via in-context learning can perform inference in dynamical systems. In particular, our goal is to examine whether standard transformers, when trained on synthetic input–output trajectories from randomly generated systems, can learn to perform filtering tasks without explicit model supervision or architectural modifications. The study is guided by constructive arguments about the representational capacity of transformers. While we do not offer convergence guarantees, our results show that with sufficient size and context, transformers can approximate Kalman-like behavior in linear settings and extend to nonlinear regimes, sometimes outperforming classical filters. Performance may degrade under limited model capacity or severely reduced context, as discussed in Section 4.

## 1.1 Related work

Prior research at the interface of deep learning and Kalman filtering include Deep Kalman Filters (Krishnan et al., 2015) and KalmanNet (Revach et al., 2021), which learns the Kalman gain in a data-driven manner using a recurrent neural network. A follow-up study (Revach et al., 2022) extends this approach using gated recurrent units to estimate both the Kalman gain and noise statistics while assuming fixed system parameters. In contrast, our model is trained on data generated from randomly sampled system parameters, encouraging the transformer to learn the filtering procedure itself rather than memorize input–output mappings for a specific system. Other works have explored connections between attention mechanisms and structured state-space models (SSMs). Dao & Gu (2024) reformulate SSM computations as matrix multiplications on

structured matrices, drawing parallels with efficient attention variants. Sieber et al. (2024) propose a unified dynamical systems framework encompassing attention, SSMs, RNNs, and LSTMs. However, neither of these works address state estimation, filtering, or in-context learning. Goel & Bartlett (2024) show that softmax self-attention can approximate the Nadaraya–Watson kernel smoother, which in turn resembles the Kalman filter. In contrast, our work directly targets in-context learning for dynamical systems and builds on the raw operator framework of Akyürek et al. (2023) to show that transformers can implement the exact operations needed for Kalman filtering, as supported by both theoretical constructions and empirical validation.

### 1.2 Summary of contributions

We present, to our knowledge, the first study demonstrating that a transformer, pre-trained on trajectories generated from randomly sampled system parameters, can in-context learn to perform filtering in dynamical systems. Specifically, we show that a frozen transformer, when conditioned on a short context of input–output pairs (and, optionally, system parameters), can predict the current output without test-time updates or direct model supervision. Our contributions are as follows:

- We provide a proof-by-construction showing that the Kalman filter can be reformulated using operations readily implementable by a transformer. Using the mean squared prediction difference (MSPD), we empirically demonstrate that a transformer can in-context learn to emulate the Kalman filter tailored to individual systems.

- Beyond linear systems, we show that transformers can in-context learn to perform accurate output prediction in certain nonlinear dynamical systems. This includes a challenging maneuvering target tracking task with unknown turning rate, where the performance is comparable to that of the Extended Kalman Filter and particle filtering.

- We evaluate robustness by withholding portions of the system model from the prompt. Notably, even in the absence of the state-transition matrix, the transformer approximates the operations and predictive accuracy of the Dual Kalman filter, demonstrating implicit parameter inference and context-level adaptability.

- We observe that transformer behavior depends on scale: Small models and short contexts tend to emulate classical regression methods (e.g., SGD, Ridge, OLS), which do not involve latent state inference, while larger models and longer contexts exhibit filtering behavior that suggests implicit recovery of hidden states, approximating Kalman, Extended Kalman, and particle filters.

The remainder of the paper is organized as follows. Section 2 reviews relevant background on in-context learning and filtering. Section 3 introduces the system model and presents constructive arguments showing that transformers can in-context learn to implement Kalman filtering under white observation noise. Section 4 presents simulation results, including applications to non-linear systems and robustness to missing model parameters. Section 5 concludes the paper. The code used to generate experimental results is available here.

## 2 Background

### 2.1 Transformers

Transformers, introduced by Vaswani et al. (2017), are a class of neural network architectures that rely on an attention mechanism to capture relationships among elements in an input sequence. This mechanism enables transformers to model long-range dependencies and has been central to their success in sequence-to-sequence tasks. The experiments in this paper utilize the GPT-2 architecture (Radford et al., 2019), a decoder-only variant of the transformer.

A brief overview of the attention mechanism sets the stage for the discussion that follows. Let $G^{(l-1)}$ denote the input to the $l^{\text{th}}$ layer. A single attention head, indexed by $\gamma$, consists of key, query, and value matrices denoted by $W_\gamma^K$, $W_\gamma^Q$, and $W_\gamma^V$, respectively. The output of head $\gamma$ is computed as

$$b_\gamma^l = \text{Softmax}\left((W_\gamma^Q G^{(l-1)})^T (W_\gamma^K G^{(l-1)})\right)\left(W_\gamma^V G^{(l-1)}\right). \tag{1}$$

The softmax matrix in equation (1) assigns attention weights that indicate how strongly each token attends to others in the sequence. The outputs of all $B$ heads are concatenated and projected via $W^F$, yielding

$$A^l = W^F[b_1^l, b_2^l, ..., b_B^l]. \tag{2}$$

This result is added to the layer input and passed through a feedforward block to produce the output of the $l^{\text{th}}$ layer

$$G^{(l)} = W_1 \sigma \left( W_2 \lambda \left( A^l + G^{(l-1)} \right) \right) + A^l + G^{(l-1)}, \tag{3}$$

where $\lambda$ denotes layer normalization and $\sigma$ is the activation function. In our experiments, we use the Gaussian Error Linear Unit (GeLU) activation (Hendrycks & Gimpel, 2016).

## 2.2 State-space models for linear dynamical systems

Linear dynamical systems can be described by a finite-dimensional state-space model involving hidden states $x_t \in \mathbb{R}^n$ and observations $y_t \in \mathbb{R}^m$, related by the equations

$$x_{t+1} = F_t x_t + q_t \tag{4}$$
$$y_t = H_t x_t + r_t, \tag{5}$$

where $q_t \in \mathbb{R}^n$ and $r_t \in \mathbb{R}^m$ denote zero-mean white process and measurement noise, respectively (for simplicity, we assume no external control input $u_t$). The state equation (4), parameterized by the state transition matrix $F_t \in \mathbb{R}^{n \times n}$ and the noise covariance matrix $Q$, models the temporal evolution of the latent state. The measurement equation (5), parameterized by the measurement matrix $H_t \in \mathbb{R}^{m \times n}$ and the noise covariance $R$, defines how observations are generated from the underlying state via a linear transformation.

State-space models are widely used across machine learning (Gu et al., 2021), computational neuroscience (Barbieri et al., 2004), control theory (Kailath, 1980), signal processing (Kailath et al., 2000), and economics (Zeng & Wu, 2013). A central problem in many of these domains is the estimation of the latent state sequence $x_t$ and its functions (e.g., the system's output) given noisy observations $y_t$ and the parameters of the model.

## 2.3 In-context learning in the absence of dynamics

When $F = I_{n \times n}$, $Q = 0$, $H = h_t \in \mathbb{R}^{1 \times n}$ (scalar measurements), and $x_0 = x$, the state space model simplifies to

$$x_t = x \tag{6}$$
$$y_t = h_t x_t + r_t, \tag{7}$$

i.e., the state becomes time-invariant and the system reduces to a linear measurement model. At the crux of estimation problems in this setting is the inference of the unknown random vector $x$ from past measurements $y_1, y_2, \ldots, y_N$ and the corresponding measurements vectors $h_1, h_2, \ldots, h_N$. This problem can be addressed using several classical methods:

- **Stochastic Gradient Descent.** Initialize with $\hat{x}_0 = 0_{n \times 1}$, and update recursively as

$$\hat{x}_t = \hat{x}_{t-1} - 2\alpha(h_{t-1}^T h_{t-1} \hat{x}_{t-1} - h_{t-1}^T y_{t-1}), \tag{8}$$

  where $\alpha$ denotes the learning rate. Once a pre-specified convergence criterion is met, the final estimate is set to $\hat{x}_{SGD} = \hat{x}_N$.

- **Ordinary Least Squares (OLS).** Form the matrix $\bar{H} \in \mathbb{R}^{N \times n}$ with rows $h_1, h_2, \ldots, h_N$, and let $\bar{Y} = [y_1, y_2, ..., y_N]^T$. The OLS estimate is

$$\hat{x}_{OLS} = (\bar{H}^T \bar{H})^{-1} \bar{H}^T \bar{Y}. \tag{9}$$

- **Ridge Regression.** To reduce overfitting, OLS can be regularized as

$$\hat{x}_{Ridge} = (\bar{H}^T \bar{H} + \lambda I_{n \times n})^{-1} \bar{H}^T \bar{Y}, \tag{10}$$

where $\lambda$ denotes the regularization parameter.

Note that if $\lambda = \frac{\sigma^2}{\tau^2}$, where $\sigma^2$ is the variance of the measurement noise and $\tau^2$ is the variance of the prior on the latent vector $x_0 = x$, then ridge regression yields the lowest mean square error among all linear estimators of $x$, i.e., those that linearly combine measurements $y_1, ..., y_N$ to form $\hat{x}$. Moreover, if $x_0 \sim \mathcal{N}(0, \tau^2 I)$ and $r_t \sim \mathcal{N}(0, \sigma^2 I)$, then the ridge regression solution coincides with the minimum mean square error (MMSE) estimate, i.e., $\hat{x} = E[X|y_1, ..., y_N]$.

A pioneering study that explored the ability of language models to learn linear functions and implement simple algorithms was presented by Garg et al. (2022). Building on that work, Akyürek et al. (2023) examined whether GPT-2–based transformers can in-context learn the setting where $x_t = x$ as in (7). Specifically, they investigated which algorithms the transformer implicitly learns to implement when tasked with predicting $y_N$, given an input formatted as the matrix

$$\begin{bmatrix} 0 & y_1 & 0 & y_2 & ... & 0 & y_{N-1} & 0 \\ h_1^T & 0 & h_2^T & 0 & ... & h_{N-1}^T & 0 & h_N^T \end{bmatrix}.$$

In Akyürek et al. (2023), the transformer was trained on batches of examples constructed from randomly sampled latent states and measurement parameters. The authors observed that the model's behavior depends on both the architecture size and the context length. For small models trained with short contexts, the transformer approximates stochastic gradient descent (SGD). For moderate context lengths (up to the dimensionality of the latent state) and moderately sized models, its behavior resembles ridge regression. Finally, with large architectures and context lengths exceeding the latent dimension, the transformer's performance approaches that of ordinary least squares (OLS).

A major contribution of Akyürek et al. (2023) was to theoretically demonstrate that transformers can approximate the operations required to implement SGD and closed-form regression. This was accomplished with the *RAW* (Read–Arithmetic–Write) operator (see Appendix G), parameterized by matrices $W_o$, $W_a$, and $W$, and with an element-wise operator $\circ \in \{+, *\}$. The RAW operator maps the input to layer $l$, denoted $G^{(l)}$, to the output $G^{(l+1)}$ using index sets $s$, $r$, $w$, a time map $K$, and token positions $i = 1, \dots, 2N$ as

$$G_{i,w}^{(l+1)} = W_o \left( \left[ \frac{W_a}{|K(i)|} \sum_{k \in K(i)} G_k^{(l)}[r] \right] \circ W G_i^{(l)}[s] \right), \tag{11}$$

$$G_{i,j \notin w}^{(l+1)} = G_{i,j \notin w}^{(l)}. \tag{12}$$

A single transformer head can approximate this operator for arbitrary $W_o$, $W_a$, $W$, and $\circ$. With specific parameterizations, it can implement primitives such as affine transformations, matrix multiplication, scalar division, dot products, and memory-based read/write operations.

## 3  In-Context Learning for Dynamical Systems

We begin by outlining an in-context learning procedure for the generic linear state-space model defined in (4)–(5), assuming a time-invariant state equation (i.e., $F_t = F \neq I$, $Q \neq 0$). For simplicity of presentation, we first consider the case of scalar measurements. In this setting, the optimal causal linear estimator of the state sequence $x_t$, in terms of mean-square error, is the well-known Kalman filter (Kalman, 1960). The procedure begins by specifying the initial state estimate and its corresponding error covariance matrix, denoted $\hat{x}_0^+$ and $\hat{P}_0^+$, respectively. (In our experiments, we initialize these as $\hat{x}_0^+ = 0$ and $\hat{P}_0^+ = I_{n \times n}$.) Subsequent estimates and covariances are computed recursively using the Kalman filter's prediction and update steps, as detailed by the expressions below.

**Prediction Step**:

$$\hat{x}_t^- = F\hat{x}_{t-1}^+ \tag{13}$$

$$\hat{P}_t^- = F\hat{P}_{t-1}^+ F^T + Q \tag{14}$$

**Update Step**:

$$K_t = \hat{P}_t^- H_t^T (H_t \hat{P}_t^- H_t^T + R)^{-1} \tag{15}$$

$$\hat{x}_t^+ = \hat{x}_t^- + K_t(y_t - H_t \hat{x}_t^-) \tag{16}$$

$$\hat{P}_t^+ = (I - K_t H_t)\hat{P}_t^- \tag{17}$$

In the case of scalar measurements, where $H_t = h_t$ (row vector) and $R = \sigma^2$ (scalar), the matrix inversion in (15) simplifies to scalar division, an operation that can be readily approximated by a single transformer head. The update equations in this scalar setting reduce to

$$\hat{x}_t^+ = \hat{x}_t^- + \frac{\hat{P}_t^- h_t^T}{h_t \hat{P}_t^- h_t^T + \sigma^2}(y_t - h_t \hat{x}_t^-) \tag{18}$$

$$\hat{P}_t^+ = (I - \frac{\hat{P}_t^- h_t^T h_t}{h_t \hat{P}_t^- h_t^T + \sigma^2})\hat{P}_t^- . \tag{19}$$

These simplified forms highlight the modular arithmetic structure of the Kalman update, which we later exploit in constructing transformer-executable analogs.

To examine whether a transformer can in-context learn to approximate the behavior of the Kalman filter, we design a training procedure based on synthetic data generated from linear dynamical systems with scalar observations. For each training instance, we randomly sample system parameters – including the state transition matrix $F$, the measurement vectors $h_1, \ldots, h_N$, the process noise covariance $Q$, and the measurement noise covariance $\sigma^2$ – along with the corresponding outputs $y_1, \ldots, y_{N-1}$. These quantities are arranged into a structured input matrix of dimensions $(n + 1) \times (2n + 2N + 1)$, formatted as

$$\begin{bmatrix} 0 & 0 & \sigma^2 & 0 & y_1 & 0 & y_2 & ... & y_{N-1} & 0 \\ F & Q & 0 & h_1^T & 0 & h_2^T & 0 & ... & 0 & h_N^T \end{bmatrix}. \tag{20}$$

The transformer, denoted by $T_\theta()$, is trained to predict the output at every alternate column starting from position $(2n + 1)^{st}$ where the first column is indexed as 0. In particular, the objective is to minimize the mean squared error over the prediction horizon using the loss function

$$\frac{1}{N} \sum_{t=1}^{N} (y_t - T_\theta(h_1, y_1, ..., h_{t-1}, y_{t-1}, h_t, F, Q, \sigma^2))^2. \tag{21}$$

This formulation explicitly tests whether the transformer can, based only on a short trajectory of input–output pairs and system parameters, learn to predict the current output in a way consistent with Kalman filtering.

As shown by Akyürek et al. (2023), there exists a parametrization of a transformer's head that can approximate the Read–Arithmetic–Write (RAW) operator defined in equations (11)–(12). Building on this, we identify a set of basic matrix operations, each implementable using the RAW operator, that are sufficient to reformulate the Kalman filter's prediction and update steps. These operations are defined over specific index subsets of the transformer's input matrix. For example, consider the system matrix $F$; the indices corresponding to $F$ in the input matrix of expression (20) are denoted by

$$I_F^{input} = \{(1, 0), (1, 1), (1, 2), ..., (1, n - 1), ..., (n, 0), (n, 1), ..., (n, n - 1)\}.$$

Further details on constructing these index sets are provided in the appendix.

To facilitate aforementioned reformulation of the Kalman filtering recursions, we define the following primitive operations:

- **Mul**$(I, J, K)$. Multiplies submatrices at index sets $I$ and $J$, and writes the result to index set $K$.

- **Div**$(I, j, K)$. Divides each element at indices in $I$ by the scalar at index $j$, and stores the result to index set $K$.

- **Aff**$(I, J, K, W_1, W_2)$. Performs the affine transformation $W_1 \cdot \mathrm{mat}(I) + W_2 \cdot \mathrm{mat}(J)$ and writes the result to $K$, where $\mathrm{mat}(I)$ and $\mathrm{mat}(J)$ denote submatrices at index sets $I$ and $J$, respectively.

- **Transpose**$(I, J)$. Computes the transpose of the matrix at $I$ and writes it to $J$.

With the primitive operations defined above, the Kalman filter recursions can be reformulated using transformer-executable instructions. To enable this, we introduce auxiliary notation for managing intermediate computations. We assume that the input to the transformer can be prepended with a matrix of identity and zero submatrices, denoted by $\mathcal{A}_{append}$. The full input is then $\mathcal{A}_{cat} = [\mathcal{A}_{append}, \mathcal{A}_{input}]$. Within $\mathcal{A}_{cat}$, we define the following index sets for intermediate variables:

- $I_{B1}$: an $n \times n$ identity submatrix;

- $I_{B2}$, $I_{B9}$: two $n \times n$ submatrices of zeros;

- $I_{B3}$: a $1 \times n$ row vector of zeros;

- $I_{B4}$, $I_{B8}$: two $n \times 1$ column vectors of zeros;

- $I_{B5}$, $I_{B6}$, $I_{B7}$: scalar zeros.

We also define the index sets $I_F$, $I_Q$, and $I_\sigma$ to refer to the locations of $F$, $Q$, and $\sigma^2$ in $\mathcal{A}_{cat}$, respectively.

These buffers provide writeable memory slots for storing the evolving Kalman variables (such as the state, covariance matrix, and intermediate expressions) through the course of the algorithm. The complete recursive implementation is detailed in Algorithm 1. A line-by-line explanation of how each transformer operation maps to a Kalman step is available in Appendix A. Note that Algorithm 1 serves as a symbolic abstraction describing how a generic transformer architecture can, *in principle*, emulate a Kalman filter using the **Mul**, **Div**, **Aff** operators that are special instances of the unified transformer primitive *RAW* operator. These derivations are not convergence or learning guarantees, but rather proofs of representability illustrating what the transformer architecture can compute in principle. The steps outlined in Appendix A illuminate this derivation in detail and walk through the mathematics behind each symbolic update. The empirical investigation presented later in this paper uses standard transformers, without any custom architectural modifications.

The above framework naturally extends to systems with vector-valued observations and uncorrelated noise. Suppose $y_t \in R^m$ and $r_t \sim \mathcal{N}(0, R)$, where $R$ is a diagonal matrix with positive entries $\sigma_1^2$, $\sigma_2^2$, $\ldots$, $\sigma_m^2$. Let $H_t$ denote the measurement matrix, $y_t^j$ denote the $j^{th}$ component of $y_t$, and $H_t^{(j)}$ denote the $j^{th}$ row of $H_t$. Then the Kalman update equations become (Kailath et al., 2000)

$$\hat{x}_t^{(1)+} = \hat{x}_t^- + \frac{\hat{P}_t^- H_t^{(1)T}}{H_t^{(1)} \hat{P}_t^- H_t^{(1)T} + \sigma_1^2} \left( y_t^{(1)} - H_t^{(1)T} \hat{x}_t^- \right) \tag{22}$$

$$\hat{P}_t^{(1)+} = \left( I - \frac{\hat{P}_t^- H_t^{(1)T} H_t^{(1)}}{H_t^{(1)} \hat{P}_t^- H_t^{(1)T} + \sigma_1^2} \right) \hat{P}_t^- \tag{23}$$

$$\hat{x}_t^{(j)+} = \hat{x}_t^{(j-1)+} + \frac{\hat{P}_t^{(j-1)+} H_t^{(j)T}}{H_t^{(j)} \hat{P}_t^{(j-1)+} H_t^{(j)T} + \sigma_j^2} \left( y_t^{(j)} - H_t^{(j)T} \hat{x}_t^{(j-1)+} \right) \quad j = 2, \ldots, m \tag{24}$$

$$\hat{P}_t^{(j)+} = \left( I - \frac{\hat{P}_t^{(j-1)+} H_t^{(j)T} H_t^{(j)}}{H_t^{(j)} \hat{P}_t^{(j-1)+} H_t^{(j)T} + \sigma_j^2} \right) \hat{P}_t^{(j-1)+} \quad j = 2, \ldots, m \tag{25}$$

$$\hat{x}_t^+ = \hat{x}_t^{(m)+} \tag{26}$$

$$\hat{P}_t^+ = \hat{P}_t^{(m)+} \tag{27}$$

While equations (13)-(17) present the standard Kalman filter in its classical recursive form, equations (22)-(27) re-express the same Kalman filter updates in an unrolled format that makes them amenable to transformer implementation. This recursive structure enables sequential updates for each measurement dimension and is readily encoded in a format similar to that in the scalar case. In particular, the in-context learning can be performed by providing to the transformer the input formatted as

$$
\begin{bmatrix}
0 & 0 & \sigma_1^2 & 0 & y_1^{(1)} & \ldots & 0 & y_{N-1}^{(1)} & 0 \\
0 & 0 & \sigma_2^2 & 0 & y_1^{(2)} & \ldots & 0 & y_{N-1}^{(2)} & 0 \\
. & . & . & . & . & . & . & . & . \\
. & . & . & . & . & . & . & . & . \\
0 & 0 & \sigma_m^2 & 0 & y_1^{(m)} & \ldots & 0 & y_{N-1}^{(m)} & 0 \\
F & Q & 0 & H_1^{(1)T} & 0 & \ldots & H_{N-1}^{(1)T} & 0 & H_N^{(1)T} \\
. & . & . & . & . & . & . & . & . \\
. & . & . & . & . & . & . & . & . \\
0 & 0 & 0 & H_1^{(m)T} & 0 & \ldots & H_{N-1}^{(m)T} & 0 & H_N^{(m)T}
\end{bmatrix} .
\tag{28}
$$

This layout enables a direct extension of the scalar Kalman filtering implementation to the multivariate setting, using the same transformer-executable primitives (e.g., `Mul`, `Div`, `Aff`, and `Transpose`).

---

**Algorithm 1:** *Formulating the KF recursions using elementary operations implementable by transformers.*

**Input:** $\mathcal{A}_{\text{cat}}, I_F, I_Q, I_\sigma, I_{B1}, I_{B2}, I_{B3}, I_{B4}, I_{B5}, I_{B6}, I_{B7}, I_{B8}, I_{B9}$

1  **Initialize** $I_{\hat{X}_{\text{Curr}}} \leftarrow (1:n,\ 2n)$ ;
2  **for** $i = 1$ **to** $N$ **do**
3     $I_{\hat{X}_{\text{next}}} \leftarrow (1:n,\ 2n+2i)$ ;
4     $I_h \leftarrow (1:n,\ 2n+2i-1)$ ;
5     $I_y \leftarrow (0,\ 2n+2i)$ ;
6     **Transpose**$(I_F,\ I_{B2})$ ;
7     **Mul**$(I_F,\ I_{\hat{X}_{\text{Curr}}},\ I_{\hat{X}_{\text{next}}})$ ;
8     **Mul**$(I_F,\ I_{B1},\ I_{B1})$ ;
9     **Mul**$(I_{B1},\ I_{B2},\ I_{B1})$ ;
10    **Aff**$(I_{B1},\ I_Q,\ I_{B1},\ W_1 = I_{n \times n},\ W_2 = I_{n \times n})$ ;
11    **Transpose**$(I_h,\ I_{B3})$ ;
12    **Mul**$(I_{B1},\ I_h,\ I_{B4})$ ;
13    **Mul**$(I_{B3},\ I_{B4},\ I_{B5})$ ;
14    **Aff**$(I_{B5},\ I_\sigma,\ I_{B6},\ W_1 = 1,\ W_2 = 1)$ ;
15    **Div**$(I_{B4},\ I_{B6},\ I_{B4})$ ;
16    **Mul**$(I_h,\ I_{\hat{X}_{\text{next}}},\ I_{B7})$ ;
17    **Aff**$(I_y,\ I_{B7},\ I_{B7},\ W_1 = 1,\ W_2 = -1)$ ;
18    **Mul**$(I_{B7},\ I_{B4},\ I_{B8})$ ;
19    **Aff**$(I_{\hat{X}_{\text{next}}},\ I_{B8},\ I_{\hat{X}_{\text{next}}},\ W_1 = 1,\ W_2 = 1)$ ;
20    **Mul**$(I_{B4},\ I_{B3},\ I_{B9})$ ;
21    **Mul**$(I_{B9},\ I_{B1},\ I_{B9})$ ;
22    **Aff**$(I_{B1},\ I_{B9},\ I_{B1},\ W_1 = I_{n \times n},\ W_2 = -I_{n \times n})$ ;
23    $I_{\hat{X}_{\text{Curr}}} \leftarrow I_{\hat{X}_{\text{next}}}$ ;

---

## 4 Simulation Results

### 4.1 Experimental setup

For transparency and reproducibility, we build upon the code and model released by Garg et al. (2022). We adopt curriculum learning, starting with a context length of $N = 10$ and incrementing it by 2 every 2000 training steps until reaching $N = 40$. In our setting, context length refers to the number of past measurements

and associated parameters available to an algorithm at inference time. Unless stated otherwise, the presented results are obtained for the hidden state dimension set to $n = 8$.

Training is performed using the Adam optimizer (Kingma, 2014) with a learning rate of 0.0001 and a batch size of 64. For each training example, $x_0$ and the measurement matrices $H_1, H_2, \ldots, H_N$ are sampled from isotropic Gaussian distributions. The process noise $q_t$ is sampled from $\mathcal{N}(0, Q)$, where $Q = U_Q \Sigma_Q U_Q^T$. Here, $U_Q$ is a randomly sampled $8 \times 8$ orthonormal matrix and $\Sigma_Q$ is diagonal with entries drawn from the uniform distribution $\mathcal{U}[0, \sigma_q^2]$. A training curriculum gradually increases $\sigma_q^2$ over 100,000 steps, after which it is held constant at 0.025. Similarly, the measurement noise $r_t$ is sampled from $\mathcal{N}(0, R)$, where $R$ is diagonal with entries $\sigma_1^2, \ldots, \sigma_m^2$ drawn from $\mathcal{U}[0, \sigma_r^2]$, with $\sigma_r^2$ also increasing over the first 100,000 training steps to a fixed value of 0.025. Both $Q$ and $R$ are re-sampled independently for each training example.

We consider two strategies for generating the state transition matrix $F$.

1. **Strategy 1 (Unitary-Interpolated Dynamics):** We generate the state transition matrix as

$$F = (1 - \alpha)I + \alpha U_F,$$

   where $\alpha \sim \mathcal{U}[0, 1]$ and $U_F$ is a randomly sampled orthonormal matrix. As a result, the eigenvalues of $F$ are generally complex and can be expressed as $pe^{j\phi}$. For a fixed $\alpha$, the phase range $\phi \in [-\phi_\alpha, \phi_\alpha]$ expands from 0 to $\pi$ as $\alpha$ increases from 0 to 1. At the endpoints $\alpha = 0$ and $\alpha = 1$, all eigenvalues lie on the unit circle; for intermediate values of $\alpha$, they may also lie inside it.

   Since $F$ may have eigenvalues on or near the unit circle, systems generated under this strategy are not necessarily stable. In fact, we empirically observe that when $\alpha = 1$, the transformer's loss fails to decrease, indicating poor convergence. To address this, we implement a training schedule in which $\alpha$ is gradually increased from 0 to 1 over 50,000 steps and then held constant.

2. **Strategy 2 (Guaranteed Stable Dynamics):** The state transition matrix is constructed as

$$F = U_F \Sigma_F U_F^T,$$

   where $U_F$ is a random orthonormal matrix and $\Sigma_F$ is diagonal with entries sampled from $\mathcal{U}[-1, 1]$. This ensures that all eigenvalues of $F$ lie strictly within the unit circle, yielding a stable system.

To evaluate the transformer's ability to approximate baseline estimators, we report both mean-squared error (MSE) and mean-squared prediction difference (MSPD). The MSE quantifies the average squared error between the transformer's predicted output and the true output. MSPD, on the other hand, compares the predictions of two models directly, regardless of ground truth. For two algorithms $\mathcal{A}_1$ and $\mathcal{A}_2$, and for a given context $\mathcal{D} = [H_1, \ldots, H_{N-1}] \sim p(\mathcal{D})$, the MSPD is defined as:

$$\text{MSPD}(\mathcal{A}_1, \mathcal{A}_2) = \mathbb{E}_{\mathcal{D}, h_N \sim p(h)} \left[ (\mathcal{A}_1(\mathcal{D})(h_N) - \mathcal{A}_2(\mathcal{D})(h_N))^2 \right]. \tag{29}$$

### 4.2 Results on linear systems

We start by evaluating the transformer in an in-context learning setting on linear dynamical systems. The transformer model used in these experiments has 32 layers, 4 attention heads and a hidden size of 512. Unless otherwise noted, evaluation is performed on batches of 5000 randomly sampled examples. The noise covariances are fixed at $\sigma_q^2 = \sigma_r^2 = 0.025$, and the Kalman filter baseline is initialized with a zero state estimate and identity error covariance.

Before analyzing in-context output prediction performance, we first demonstrate that transformers can explicitly estimate the latent state sequence from scalar observations provided in the format of expression (20). To this end, we train a transformer under the unitary-interpolated dynamics regime (Strategy 1) to directly predict the hidden state using the loss function

$$\frac{1}{N} \sum_{t=1}^{N} (x_t - T_\theta(h_1, y_1, ..., h_{t-1}, y_{t-1}, h_t, F, Q, \sigma^2))^2. \tag{30}$$

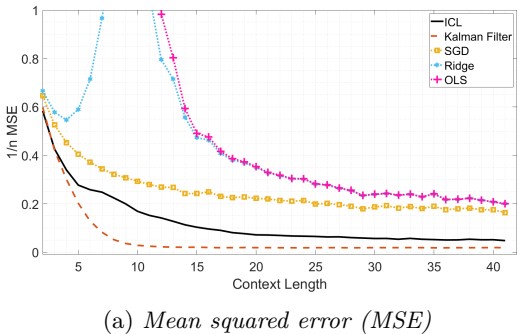 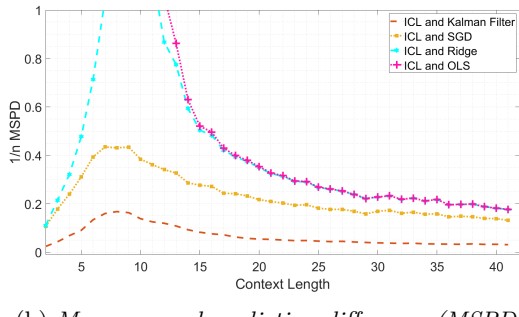

(a) *Mean squared error (MSE)*    (b) *Mean-squared prediction difference (MSPD)*

Figure 1: *Performance of the transformer in explicit state estimation from scalar measurements (Strategy 1). The transformer's predictions are compared to Kalman filtering, SGD, ridge regression, and OLS.*

Figure 1(a) shows the mean-squared error (MSE) between the transformer's prediction of $x_N$ and the ground truth, compared against several baselines including Kalman filtering, ordinary least squares, ridge regression ($\lambda = 0.05$), and stochastic gradient descent (with learning rate of $\alpha = 0.01$). The rationale behind these parameter choices is detailed in Appendix E.3. The corresponding MSPD values between transformer predictions and each baseline are reported in Figure 1(b). The Kalman filter achieves the lowest MSE, as expected. As context length increases, the transformer's performance converges to that of the Kalman filter, distancing itself away from the regression-based estimators which do not account for temporal dynamics. A notable trend observed for both ridge regression and SGD baselines is a sharp rise in MSE at small-to-intermediate context lengths, followed by recovery at longer horizons. This behavior, recurring across many experiments, is consistent with the double descent phenomenon (Schaeffer et al., 2023) wherein models initially transition from an over-parameterized to an under-parameterized regime. When the number of input features (i.e., context length) grows beyond the model's capacity to disambiguate noise from signal, overfitting leads to increased test error. As context continues to grow, the model eventually gains sufficient statistical leverage to suppress noise and recover performance. This effect is more pronounced for OLS, while the degradation in Ridge regression is milder due to the stabilizing effect of regularization.

We next assess the transformer's ability to perform in-context one-step output prediction. Given a context of past inputs and outputs, the model is tasked with predicting the system output $y_t$ at each step. The resulting MSE and MSPD for Strategy 1 are shown in Figures 2(a) and 2(b), respectively; their counterparts for Strategy 2 are shown in Figures 2(c) and  2(d). Under Strategy 1, performance of the transformer is closest to that of the Kalman filter. Under Strategy 2, at short context lengths, the transformer's predictions most closely resemble those of stochastic gradient descent (SGD) with learning rate 0.01. As the context length increases, its behavior progressively aligns with that of the Kalman filter. The differences between methods are more pronounced under Strategy 1, where the system may be unstable, and narrower under Strategy 2, where the dynamics are constrained to be stable.

To assess the robustness of the transformer to incomplete context, we repeat the previous experiment while withholding the noise covariance matrices $R$ and $Q$. As shown in Figure 3, the transformer's performance remains stable in both strategies, with minimal degradation in MSE or MSPD. These findings suggest that the transformer may be implicitly inferring the missing noise statistics as part of its in-context learning process. Note that the Kalman filter baseline still has access to the full model parameters, including noise covariances. If this information were withheld from the Kalman filter as well, estimation would require a more complex approach such as expectation-maximization (EM) to recover the unknown covariances.

We next evaluate the transformer's ability to perform in-context learning in systems with multi-dimensional (non-scalar) measurements. Specifically, we consider observations of dimension two and white Gaussian noise. The transformer's input is formatted according to expression (28), and includes the full parameterization of the state-space model. Figure 4 shows both the mean-squared error (left) and the mean-squared prediction difference (right) between the transformer's predictions and those of several baselines. The results confirm that the transformer emulates the Kalman filter in this more general setting.

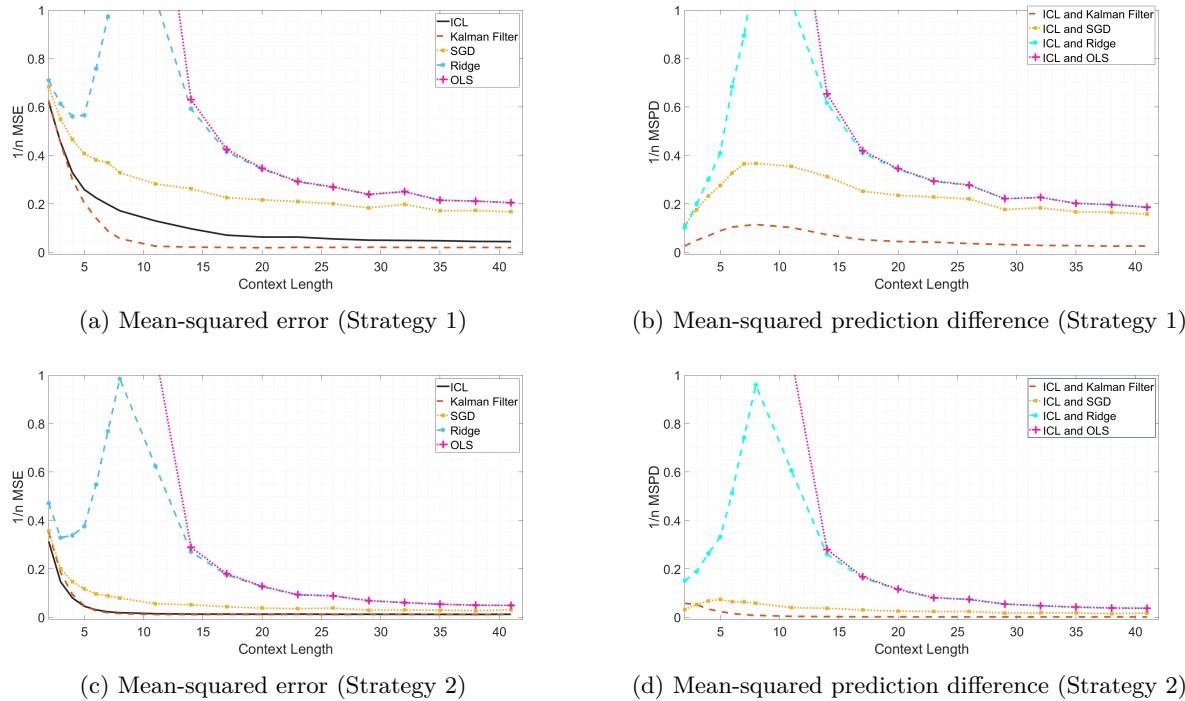

(a) Mean-squared error (Strategy 1)

(b) Mean-squared prediction difference (Strategy 1)

(c) Mean-squared error (Strategy 2)

(d) Mean-squared prediction difference (Strategy 2)

Figure 2: *Comparison between transformer and classical estimators in one-step output prediction for scalar measurements.*

Our final set of experiments involving linear dynamical systems investigates the transformer's ability to perform in-context learning when no model parameters (i.e., neither the state transition matrix nor the noise covariances) are provided in the context. To keep the setup minimal, we revert to scalar measurements. Figure 5 presents both the MSE and MSPD of the transformer relative to various baselines. As seen in the figure, when faced with the challenging task of both capturing the state dynamics and implicitly estimating unknown state transition matrix, the transformer struggles and achieves performance that most closely resembles performance of SGD. However, if this task is rendered somewhat easier by reducing dimension of the state vector from $n = 8$ to $n = 2$, thus consequently reducing the number of parameters needed to specify unknown transition matrix, the same transformer model succeeds in closely emulating performance of the Kalman filter that is provided the information withheld from the transformer. Remarkably, despite the absence of model-specific information, the transformer's performance progressively approaches that of the Kalman filter as the context length increases. This behavior is reminiscent of the Dual Kalman Filter (DKF)(Wan & Nelson, 1996), which alternates between estimating the hidden state and the unknown state transition matrix. Specifically, the transition matrix is treated as a latent variable and estimated via a secondary Kalman filter that uses a regressor constructed from prior state estimates. Appendices B and C present theoretical arguments and an implementation blueprint showing how a transformer can emulate the DKF in this setting.

Appendix E.1 presents results evaluating the transformer's robustness to distribution shifts between training and inference. For instance, a model trained under Strategy 1 with measurement matrices $H_i$ drawn from $\mathcal{N}(0, 1)$ is tested on systems where $H_i \sim \mathcal{U}[0, 3]$. Despite this substantial change in the observation model, the transformer's MSPD relative to the Kalman filter remains low, indicating strong generalization. Appendix E further evaluates robustness to variations in state dimensionality. Finally, Appendix D demonstrates that the transformer's in-context learning capabilities extend naturally to systems with control inputs, and provides implementation details and input formatting for this setting.

**Remark 1:** While in all of the reported figures the MSE and MSPD curves appear to flatten beyond context length $\approx$30, both metrics continue to decrease, albeit at a slower rate. This diminishing improvement reflects

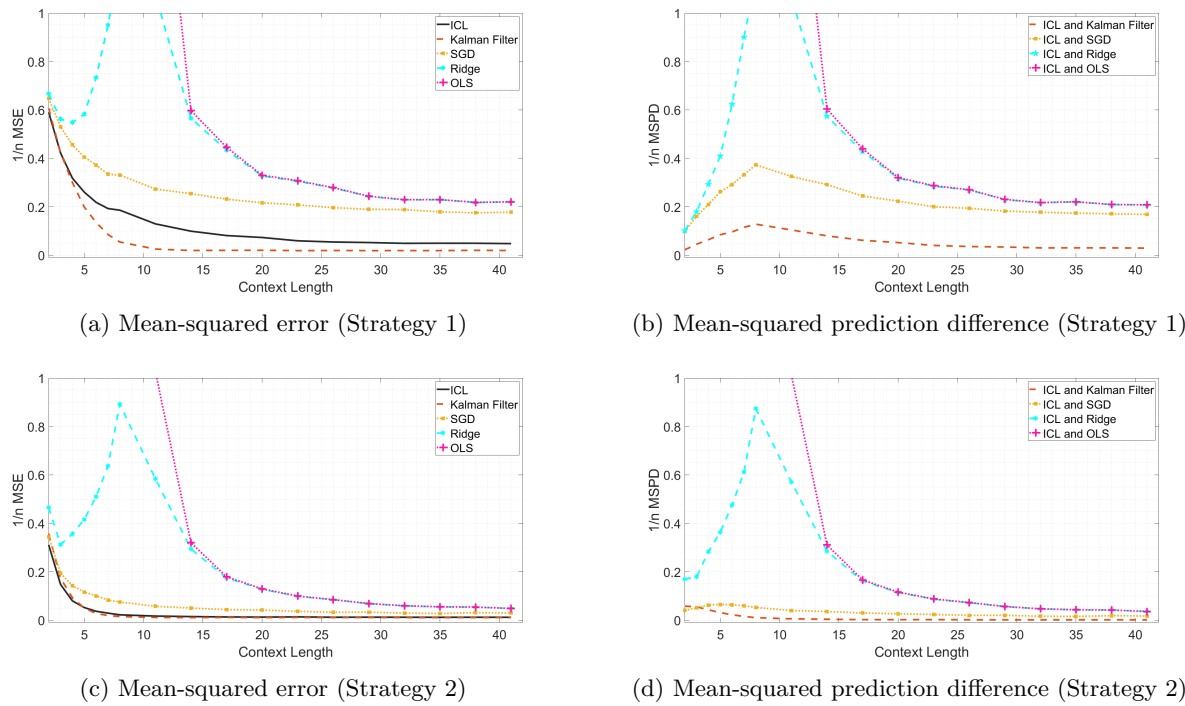

(a) Mean-squared error (Strategy 1)

(b) Mean-squared prediction difference (Strategy 1)

(c) Mean-squared error (Strategy 2)

(d) Mean-squared prediction difference (Strategy 2)

Figure 3: *Performance of in-context learning with a transformer that is not provided noise covariances $Q$ and $R$. (a) Mean-square error (MSE) under Strategy 1. (b) Mean-squared prediction difference (MSPD) relative to the baselines under Strategy 1. (c) MSE under Strategy 2. (d) MSPD relative to the baselines under Strategy 2.*

the interaction between context length and model expressivity. As context increases, the transformer has more signal to infer the latent structure, but unless the model architecture is sufficiently expressive, its predictions may remain biased relative to the Kalman filter. This behavior is further studied in nonlinear systems settings and corroborated by the results reported in Tables 1 and 2 of Section 4.3, which show that beyond a certain point, error reduction is limited not by context length but by the transformer's architectural capacity to emulate optimal inference.

**Remark 2:** We emphasize that multi-step prediction is feasible in our setup by recursively feeding the predicted latent state back into the input stream. While this introduces some accumulation of prediction error, our results in Appendix E.4 demonstrate that the transformer maintains strong performance and closely emulates the Kalman filter at horizons of up to 5 steps.

### 4.3 Results on nonlinear systems

We now assess whether transformers can in-context learn to emulate filtering behavior in nonlinear dynamical systems. As discussed in Appendix F, key operations of the Extended Kalman Filter (EKF) can be implemented using transformer primitives, suggesting that transformers are capable of learning nonlinear filtering strategies via in-context learning (ICL). To further contextualize performance, we also compare it against particle filtering (PF), which is widely used for nonlinear systems with non-Gaussian posteriors. Note that PF relies on stochastic sampling and resampling steps, which do not lend themselves as naturally to transformer architectures.

We compare the performance of transformer with baseline algorithms on two representative systems:

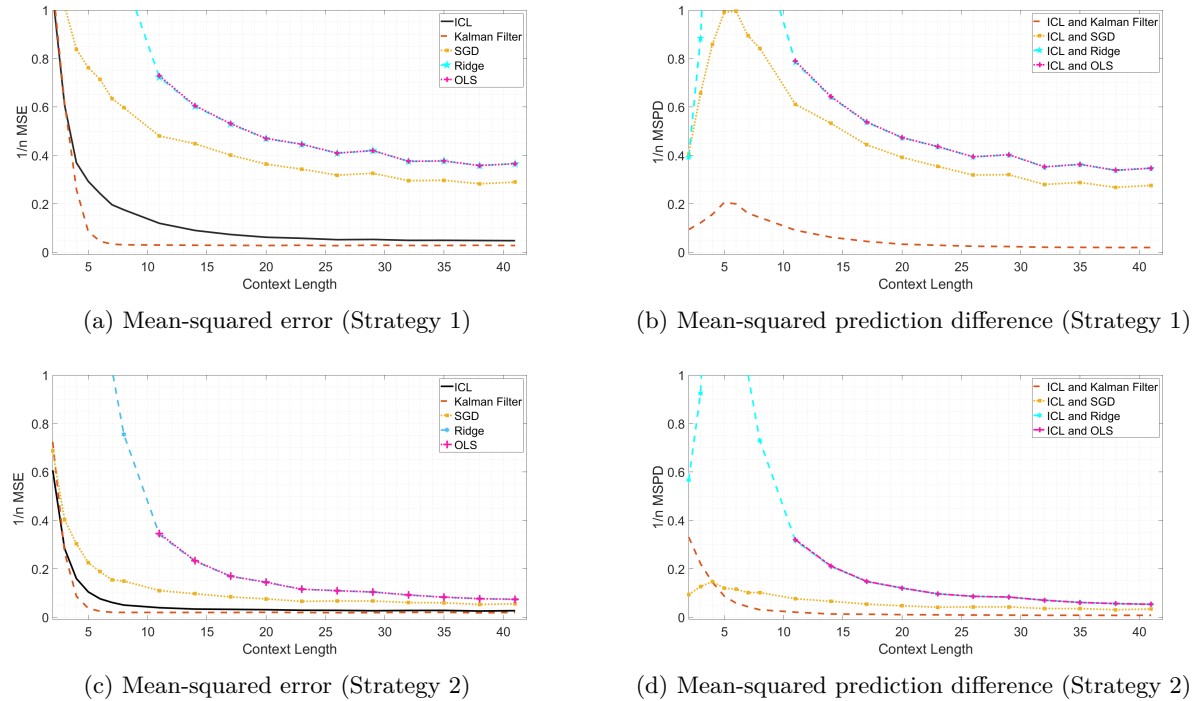

Figure 4: *In-context learning (ICL) with a transformer for systems with 2D measurements. The transformer receives the full system specification and performs one-step output prediction.*

1. **System 1 (Nonlinear State Evolution)**: A system with nonlinear state dynamics and linear measurements:

$$x_{t+1} = F \tanh(2x_t) + q_t \tag{31}$$

$$y_t = H_t x_t + r_t. \tag{32}$$

Here, the state and measurement dimensions are fixed to 2. The matrix $F$ is given by $F = 0.8I + 0.15U_F$, where $U_F$ has entries drawn from a standard Gaussian distribution. The process noise covariance is set to $Q = 0.01(I + 0.1Z_q)$, with $Z_q$ also drawn from a standard normal distribution. The observation noise covariance is $R = 0.01I$, and the measurement matrices $H_t$ are sampled from an isotropic Gaussian distribution. Variations of this system with different state transition functions and dimensionalities are explored in Appendix F.

2. **System 2 (Maneuvering Target with Unknown Turn Rate)**: A nonlinear tracking model with time-varying dynamics and measurement equations. Full model details are provided in Appendix F.

The transformer model used to obtain results in Figures 6- 7 has 16 layers, 4 attention heads and a hidden size of 512. Figure 6 reports the mean-squared error (MSE) and mean-squared prediction difference (MSPD) for System 1, where the variance of the process and measurement noise is $\sigma_q^2 = \sigma_r^2 = 0.0125$. The results demonstrate that the transformer closely tracks the performance of both the EKF and PF, indicating that it learns to emulate nonlinear filtering behavior in this regime.

Figure 7 presents results for System 2, a more complex maneuvering target model with nonlinear observations. Interestingly, in this setting the transformer achieves the best performance (i.e., the lowest MSE) among all the considered methods. This indicates that in-context learning generalizes effectively even when key latent parameters, such as turn rate, are unobserved.[1] Linear regression and SGD are omitted from this comparison due to the nonlinearity of the measurement model.

---

[1]Note that EKF and particle filter incorporate the turn rate into the state vector and thus estimate it recursively – for details, please see Appendix F.2.

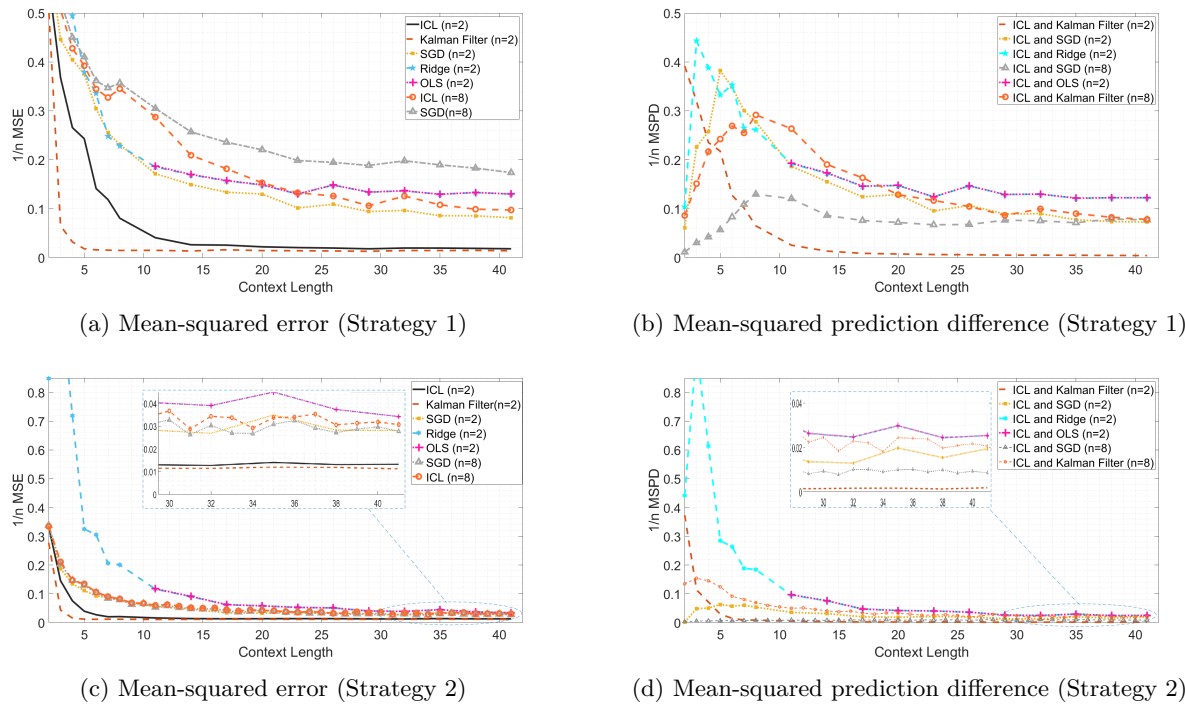

(a) Mean-squared error (Strategy 1)  (b) Mean-squared prediction difference (Strategy 1)

(c) Mean-squared error (Strategy 2)  (d) Mean-squared prediction difference (Strategy 2)

Figure 5: *Performance of in-context learning (ICL) with a transformer under fully missing model parameters (scalar measurements). No information about the state transition matrix or noise covariances is included in the context.*

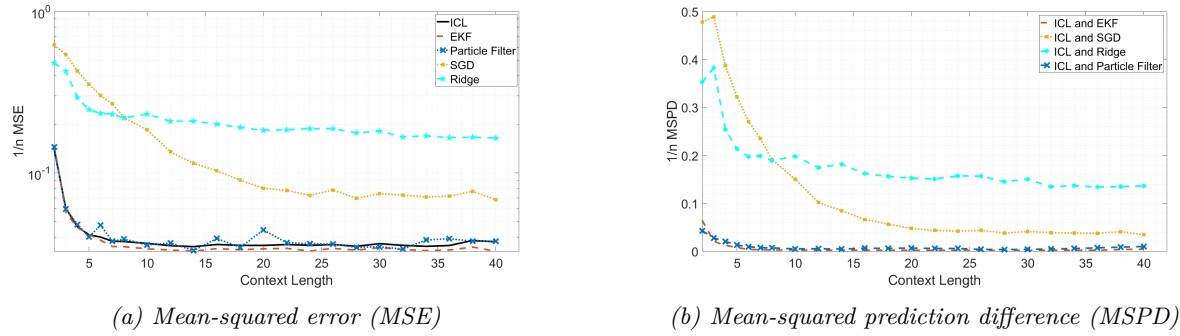

*(a) Mean-squared error (MSE)*  *(b) Mean-squared prediction difference (MSPD)*

Figure 6: *Performance of in-context learning (ICL) on System 1 with nonlinear state evolution and linear measurements. Transformer achieves performance on par with Extended Kalman Filter (EKF) and Particle Filter (PF).*

Finally, we fix the context length to 40 and investigate how the transformer's behavior in System 1 depends on model capacity, varying both the number of layers and the embedding dimension. The results, summarized in Tables 1 and 2, reveal that smaller transformers exhibit behavior more similar to that of SGD and Ridge Regression, while larger models yield outputs that increasingly align with those of EKF and the particle filter, i.e., methods that incorporate knowledge of the system's dynamical structure. This transition underscores the role of model capacity in enabling the transformer to approximate sophisticated recursive inference algorithms. Notably, even modest increases in either depth or embedding dimension significantly improve the transformer's alignment with classical filters, as measured by MSPD. These findings suggest that architectural scale is a key factor in unlocking in-context learning of dynamics-aware estimation procedures. In short,

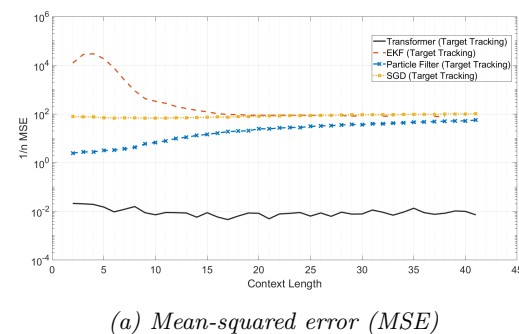

*(a) Mean-squared error (MSE)*

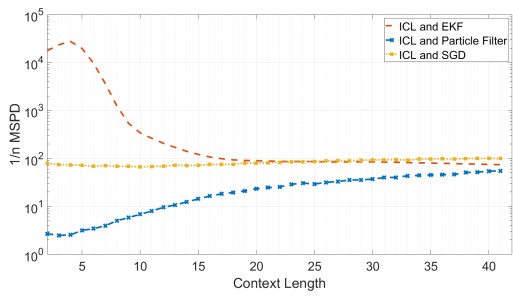

*(b) Mean-squared prediction difference (MSPD)*

Figure 7: *Performance of in-context learning (ICL) on System 2, a nonlinear tracking model with unknown turn rate. The transformer consistently outperforms particle filtering and other methods, especially at longer horizons, demonstrating its ability to handle uncertainty and perform nonlinear inference in an in-context fashion.*

while the operations required for filtering are representable in principle, sufficient model capacity is necessary to learn them reliably from data.

| Number of Layers | ICL and EKF | ICL and Particle Filter | ICL and SGD 0.01 | ICL and SGD 0.05 | ICL and Ridge 0.01 | ICL and Ridge 0.05 |
|---|---|---|---|---|---|---|
| 1 | 1.0281 | 0.9904 | **0.2662** | 1.0010 | 0.7484 | 0.7463 |
| 2 | 0.2447 | 0.2274 | 0.2433 | 0.1806 | 0.1691 | **0.1689** |
| 4 | 0.1251 | **0.1078** | 0.3598 | 0.1800 | 0.2448 | 0.2446 |
| 8 | 0.0526 | **0.0337** | 0.3970 | 0.1924 | 0.2892 | 0.2891 |
| 16 | 0.0531 | **0.0336** | 0.4273 | 0.2199 | 0.3191 | 0.3189 |

Table 1: Effects of transformer depth on MSPD in System 1. The table reports mean-squared prediction difference (MSPD) between the transformer and various baselines as the number of transformer layers increases (embedding dimension fixed to 512).

| Embedding Dim | ICL and EKF | ICL and Particle Filter | ICL and SGD 0.01 | ICL and SGD 0.05 | ICL and Ridge 0.01 | ICL and Ridge 0.05 |
|---|---|---|---|---|---|---|
| 8 | 1.068 | 1.041 | **0.283** | 1.036 | 0.795 | 0.793 |
| 32 | 0.152 | 0.130 | 0.314 | **0.126** | 0.214 | 0.214 |
| 64 | 0.104 | **0.087** | 0.366 | 0.170 | 0.264 | 0.263 |
| 256 | 0.060 | **0.046** | 0.411 | 0.215 | 0.310 | 0.310 |
| 512 | 0.053 | **0.034** | 0.397 | 0.192 | 0.289 | 0.289 |

Table 2: Effect of embedding dimension on MSPD in System 1. The table reports MSPD between the transformer and various baselines as the embedding dimension increases (number of transformer layers fixed at 8).

## 5 Conclusion

This work investigated whether transformers, trained via in-context learning on synthetic trajectories from randomly sampled dynamical systems, can implicitly learn to perform filtering without test-time gradient updates or explicit access to model equations. We provided constructive arguments showing that the Kalman filter can be expressed using operations natively supported by transformer architectures, and empirically demonstrated that, in linear systems, transformer predictions closely track those of the Kalman filter when given sufficient context and model capacity. The nature of the learned algorithm was found to depend on both model scale and context length – smaller models or shorter contexts emulate linear regression or stochastic gradient descent, while larger models with longer contexts converge toward optimal filtering behavior.

Beyond linear systems, we demonstrated that transformers can generalize to nonlinear dynamical systems as well. In particular, we showed that they can match the performance of the Extended Kalman Filter and the particle filter, and in some settings even outperform them. These results suggest that transformers do

not merely replicate a fixed algorithm but instead learn a flexible, data-driven inference strategy through in-context learning.

The robustness of transformer-based filtering to missing context, such as unobserved noise covariances or system parameters, further highlights its potential as a general-purpose alternative to manually designed filters. However, performance degrades under limited model capacity and/or missing inputs, emphasizing the importance of expressiveness and informative prompts. Future work will explore extensions to temporally correlated noise and examine the emergence of internal representations that support in-context filtering.

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

# A    Details regarding the implementation of Algorithm 1

We elaborate on the operations in Algorithm 1, focusing for simplicity on the case of scalar measurements. Let the state dimension be $n$. Define the auxiliary matrix

$$\mathcal{A}_{append} = \begin{bmatrix} B_1 & B_2 & B_9 & B_3^T & B_4 & B_8 & 0_{n\times 1} & 0_{n\times 1} & 0_{n\times 1} \\ 0_{1\times n} & 0_{1\times n} & 0_{1\times n} & 0 & 0 & 0 & B_5 & B_6 & B_7 \end{bmatrix}. \tag{33}$$

For illustration, consider $n = 2$, where

$$\mathcal{A}_{append} = \begin{bmatrix} 1 & 0 & 0 & 0 & 0 & 0 & 0 & 0 & 0 & 0 & 0 & 0 \\ 0 & 1 & 0 & 0 & 0 & 0 & 0 & 0 & 0 & 0 & 0 & 0 \\ 0 & 0 & 0 & 0 & 0 & 0 & 0 & 0 & 0 & 0 & 0 & 0 \end{bmatrix}. \tag{34}$$

Form the concatenated matrix $\mathcal{A}_{cat} = [\mathcal{A}_{append} \ \mathcal{A}_{input}]$, and define index sets such as $I_{B1} = \{(0,0),(1,0),(0,1),(1,1)\}$, $I_{B2} = \{(0,2),(1,2),(0,3),(1,3)\}$, and so on. These index sets specify memory regions corresponding to the working blocks in the transformer's computation, where operations like matrix multiplications, transpositions, and affine combinations are carried out.

To build intuition, we walk through the first iteration of the FOR loop in Algorithm 1. We initialize $\hat{x}_0^+ = 0$ and carry out a sequence of transformer-readable operations:

1. **Initialization:** Set $I_{\hat{X}_{\text{Curr}}} \leftarrow (1:n, 2n)$.

2. **Pointer setup:** For $i = 1$, set $I_{\hat{X}_{\text{next}}} \leftarrow (1:n, 2n+2)$.

3. **Measurement access:** Define
   - $I_h \leftarrow (1:n, 2n+1)$ (for $h_1^T$)
   - $I_y \leftarrow (0, 2n+2)$ (for $y_1$)

4. **Transpose:** Write $F^T$ to $B_2$ using **Transpose**$(I_F, I_{B2})$.

5. **State prediction:** Compute $\hat{x}_1^- = F\hat{x}_0^+$ via **Mul**$(I_F, I_{\hat{X}_{\text{Curr}}}, I_{\hat{X}_{\text{next}}})$.

6. **Covariance prediction:**
   - Compute $F\hat{P}_0^+$ to $B_1$
   - Multiply by $F^T$ to get $F\hat{P}_0^+ F^T$ in $B_1$
   - Add $Q$ via **Aff**$(I_{B1}, I_Q, I_{B1})$ to form $\hat{P}_1^-$

7. **Kalman gain computation:**
   - Transpose $h_1^T$ to $B_3$
   - Compute $\hat{P}_1^- h_1^T$ to $B_4$
   - Compute scalar $s = h_1 \hat{P}_1^- h_1^T$ to $B_5$
   - Add measurement noise $\sigma^2$: **Aff**$(I_{B5}, I_\sigma, I_{B6})$
   - Divide to compute $K_1$: **Div**$(I_{B4}, I_{B6}, I_{B4})$

8. **Measurement update:**
   - Compute predicted observation $h_1 \hat{x}_1^-$ to $B_7$
   - Subtract from $y_1$ to get residual in $B_7$
   - Multiply $K_1$ with residual to get $B_8$
   - Update $\hat{x}_1^+ = \hat{x}_1^- + K_1(y_1 - h_1 \hat{x}_1^-)$ to $I_{\hat{X}_{\text{next}}}$

9. **Covariance update:**
   - Compute outer product $h_1 K_1$ to $B_9$

- Multiply with $\hat{P}_1^-$ to form $h_1 K_1 \hat{P}_1^-$
- Subtract from $\hat{P}_1^-$ to form $\hat{P}_1^+$ in $B_1$

10. **Pointer update:** Set $I_{\hat{X}_{\text{Curr}}} \leftarrow I_{\hat{X}_{\text{next}}}$.

Each of these steps is directly implementable via a composition of the primitive operations defined in the main text: **Mul**, **Div**, **Aff**, and **Transpose**.

## B   A brief summary of the Dual Kalman Filter

Consider the state-space model

$$x_{t+1} = F_t x_t + q_t \tag{35}$$
$$y_t = H_t x_t + r_t, \tag{36}$$

where both the latent state $x_t$ and the transition matrix $F_t$ must be estimated. The Dual Kalman Filter (DKF), introduced by Wan & Nelson (1996), addresses this by alternating between two Kalman filtering recursions – one for $x_t$ given an estimate of $F_t$, and one for $F_t$ given an estimate of $x_t$.

Let $f_t \in \mathbb{R}^{n^2}$ be the vectorized form of $F_t$, and define the matrix $X_t \in \mathbb{R}^{n \times n^2}$ as:

$$X_t = \begin{bmatrix} \hat{x}_{t-1}^{+T} & 0 & \dots & 0 \\ 0 & \hat{x}_{t-1}^{+T} & \dots & 0 \\ \vdots & \vdots & \ddots & \vdots \\ 0 & 0 & \dots & \hat{x}_{t-1}^{+T} \end{bmatrix}. \tag{37}$$

This formulation enables the re-expression of the system as a linear model in $f_t$:

$$f_t = f_{t-1}, \tag{38}$$
$$y_t = H_{f,t} f_{t-1} + r_{f,t}, \tag{39}$$

where $H_{f,t} = H_t X_t$, and $r_{f,t} = H_t q_t + r_t$, with $r_{f,t} \sim \mathcal{N}(0, R_f)$ and $R_f = H_t Q H_t^T + R$.

The DKF then proceeds with standard Kalman filter prediction and update steps applied to $f_t$.

**Prediction Step**:

$$\hat{f}_t^- = \hat{f}_{t-1}^+ \tag{40}$$
$$\hat{P}_{f,t}^- = \hat{P}_{f,t-1}^+ \tag{41}$$

**Update Step**:

$$K_{f,t} = \hat{P}_{f,t}^- H_{f,t}^T (H_{f,t} \hat{P}_{f,t}^- H_{f,t}^T + R_f)^{-1} \tag{42}$$
$$\hat{f}_t^+ = \hat{f}_t^- + K_{f,t}(y_t - H_{f,t} \hat{f}_t^-) \tag{43}$$
$$\hat{P}_{f,t}^+ = (I - K_{f,t} H_{f,t}) \hat{P}_{f,t}^- \tag{44}$$

In the scalar measurement case, $H_{f,t} \hat{P}_{f,t}^- H_{f,t}^T$, $H_t Q H_t^T$, and $R$ reduce to scalars. Thus, the Kalman gain simplifies to

$$K_{f,t} = \frac{1}{H_{f,t} \hat{P}_{f,t}^- H_{f,t}^T + R_f} \hat{P}_{f,t}^- H_{f,t}^T. \tag{45}$$

## C  Transformer Can In-Context Learn to Perform Dual Kalman Filtering for a System with Scalar Measurements

We now consider the setting where the state transition matrix is not provided as part of the context. Let the transformer's input be

$$\begin{bmatrix} 0 & \sigma^2 & 0 & y_1 & 0 & y_2 & ... & y_{N-1} & 0 \\ Q & 0 & h_1^T & 0 & h_2^T & 0 & ... & 0 & h_N^T \end{bmatrix}. \tag{46}$$

One can argue analogously to the proof-by-construction used for the Kalman filter that a transformer can learn to perform Dual Kalman Filtering in context, even in the absence of explicit dynamics. To support this, we extend the previously introduced set of elementary operations with a new one, **MAP**$(I, J)$, which takes the vector at indices $I$ and transforms it into a matrix of the form shown in expression (37), storing the result at indices $J$.

In addition to the index sets defined in the main text (i.e., $\mathcal{A}_{cat}$, $I_F$, $I_Q$, $I\sigma$, $I_{B1}$ through $I_{B9}$), we introduce the following index sets: $I_{B10}$: indices of an $n \times n^2$ block initialized to zeros; $I_{B11}$: indices of a $1 \times n^2$ vector block; $I_{B12}$ and $I_{B14}$: indices of $n^2 \times n^2$ matrices; $I_{B13}$ and $I_{\hat{f}_{next}}$: indices of $n^2 \times 1$ vectors.

The complete Dual Kalman Filter recursion can now be expressed using operations implementable by transformers; the pseudo-code below shows one iteration of the resulting algorithm.

1. **Initialization:** Set $I_{\hat{X}_{\text{Curr}}} \leftarrow (1:n, 2n)$.

2. **Pointer setup for iteration $i = 1$:**
   - $I_{\hat{X}_{\text{next}}} \leftarrow (1:n, 2n+2)$
   - $I_h \leftarrow (1:n, 2n+1)$
   - $I_y \leftarrow (0, 2n+2)$

3. **State prediction:**
   - **Transpose**$(I_F, I_{B2})$: Write $F^T$ to $B_2$
   - **Mul**$(I_F, I_{\hat{X}_{\text{Curr}}}, I_{\hat{X}_{\text{next}}})$: Compute $\hat{x}_1^- = F\hat{x}_0^+$

4. **Covariance prediction:**
   - **Mul**$(I_F, I_{B1}, I_{B1})$: $F\hat{P}_0^+$
   - **Mul**$(I_{B1}, I_{B2}, I_{B1})$: $F\hat{P}_0^+ F^T$
   - **Aff**$(I_{B1}, I_Q, I_{B1}, W_1 = I, W_2 = I)$: $\hat{P}_1^-$

5. **Kalman gain computation:**
   - **Transpose**$(I_h, I_{B3})$: $h_1$
   - **Mul**$(I_{B1}, I_h, I_{B4})$: $\hat{P}_1^- h_1^T$
   - **Mul**$(I_{B3}, I_{B4}, I_{B5})$: $h_1 \hat{P}_1^- h_1^T$
   - **Aff**$(I_{B5}, I_\sigma, I_{B6}, W_1 = 1, W_2 = 1)$: $s = h_1 \hat{P}_1^- h_1^T + \sigma^2$
   - **Div**$(I_{B4}, I_{B6}, I_{B4})$: Kalman gain $K_1$

6. **Measurement update:**
   - **Mul**$(I_h, I_{\hat{X}_{\text{next}}}, I_{B7})$: predicted $y_1$
   - **Aff**$(I_y, I_{B7}, I_{B7}, W_1 = 1, W_2 = -1)$: residual $y_1 - \hat{y}_1$
   - **Mul**$(I_{B7}, I_{B4}, I_{B8})$: $K_1(y_1 - \hat{y}_1)$
   - **Aff**$(I_{\hat{X}_{\text{next}}}, I_{B8}, I_{\hat{X}_{\text{next}}}, W_1 = 1, W_2 = 1)$: $\hat{x}_1^+$

7. **Covariance update:**

- **Mul**($I_{B4}$, $I_{B3}$, $I_{B9}$): $h_1 K_1$
- **Mul**($I_{B9}$, $I_{B1}$, $I_{B9}$): $h_1 K_1 \hat{P}_1^-$
- **Aff**($I_{B1}$, $I_{B9}$, $I_{B1}$, $W_1 = I$, $W_2 = -I$): $\hat{P}_1^+$

8. **DKF step: latent transition matrix estimation**

- **MAP**($I_{\hat{X}_{\text{next}}}$, $I_{B10}$): Build $X_1$
- **Mul**($I_{B3}$, $I_{B10}$, $I_{B11}$): $H_{f,1} = h_1 X_1$
- **Transpose**($I_{B11}$, $I_{B13}$)
- **Mul**($I_{B11}$, $I_{B12}$, $I_{B11}$): $H_{f,1} \hat{P}_{f,1}^-$
- **Mul**($I_{B11}$, $I_{B13}$, $I_{B5}$): $H_{f,1} \hat{P}_{f,1}^- H_{f,1}^T$
- **Aff**($I_{B5}$, $I_\sigma$, $I_{B6}$, $W_1 = 1$, $W_2 = 1$): add $R_f$
- **Div**($I_{B13}$, $I_{B6}$, $I_{B13}$): Kalman gain for $f$
- **Mul**($I_{B7}$, $I_{B13}$, $I_{B8}$)
- **Aff**($I_{\hat{f}_{\text{next}}}$, $I_{B8}$, $I_{\hat{f}_{\text{next}}}$, $W_1 = 1$, $W_2 = 1$): $\hat{f}_1^+$
- **Mul**($I_{B3}$, $I_{B10}$, $I_{B11}$)
- **Mul**($I_{B13}$, $I_{B11}$, $I_{B14}$)
- **Mul**($I_{B14}$, $I_{B12}$, $I_{B14}$)
- **Aff**($I_{B12}$, $I_{B14}$, $I_{B12}$, $W_1 = I$, $W_2 = -I$): $\hat{P}_{f,1}^+$
- **MAP**($I_{\hat{f}_{\text{next}}}$, $I_F$): overwrite $F$ for next round

9. **Pointer update:** Set $I_{\hat{X}_{\text{Curr}}} \leftarrow I_{\hat{X}_{\text{next}}}$.

# D   Experiments with Control Input

The filtering setup studied in the main text can be extended to systems with control inputs, provided that the measurement noise remains white. In this case, the state-space model becomes

$$x_{t+1} = F_t x_t + B_t u_t + q_t \tag{47}$$

$$y_t = H_t x_t + r_t, \tag{48}$$

and the Kalman filter prediction step is modified accordingly:

$$\hat{x}_t^- = F\hat{x}_{t-1}^+ + Bu_t. \tag{49}$$

The derivation showing that transformers can implement Kalman filtering operations under this model follows the same constructive arguments presented for the zero-input case and is omitted for brevity. Instead, we report empirical results for scalar measurements and nonzero control inputs.

In these experiments, the control matrix $B \in \mathbb{R}^{8 \times 8}$ is generated as $B = U_B \Sigma_B U_B^T$, where $U_B$ is a random orthonormal matrix and $\Sigma_B$ is a diagonal matrix with entries sampled from $\mathcal{U}[-1, 1]$. Control vectors $u_t \in \mathbb{R}^8$ are sampled from a standard Gaussian distribution and then normalized to unit norm.

The input to the transformer in this setting is formatted as

$$\begin{bmatrix} 0 & 0 & 0 & \sigma^2 & 0 & 0 & 0 & y_1 & \cdots & 0 & 0 \\ F & Q & B & 0 & 0 & h_1^T & u_1 & 0 & \cdots & h_N^T & u_N \end{bmatrix}. \tag{50}$$

Here, $F$ is generated using the Unitary-Interpolated Dynamics strategy (Strategy 1), and the remaining settings match those of the baseline experiments without control.

Figure 8 shows the results. The transformer maintains low MSE relative to the ground truth and exhibits MSPD behavior similar to that of the Kalman filter, indicating successful in-context learning in the presence of control inputs.

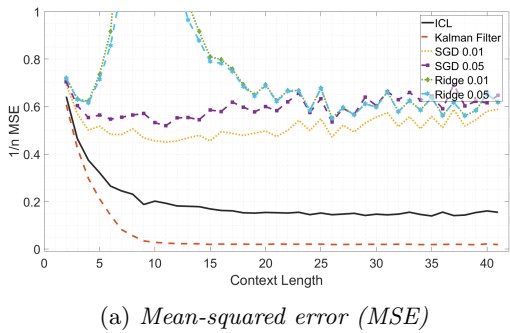 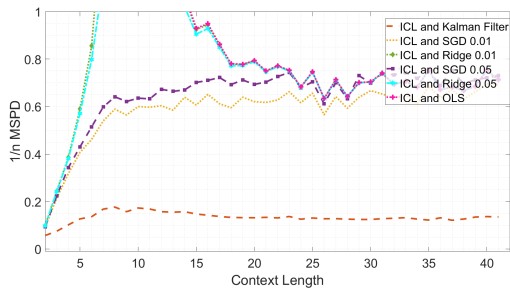

(a) *Mean-squared error (MSE)*          (b) *Mean-squared prediction difference (MSPD)*

Figure 8: *Transformer performance on linear systems with control inputs. The model retains effective filtering behavior even with added input terms.*

# E    Additional Experiments and Further Details

## E.1    An illustration of the performance on out-of-sample parameters

To evaluate the performance of the transformer on systems with parameters sampled from a distribution different from that used during training, we train the model with $F$ generated using Strategy 1 and $H_i \sim \mathcal{N}(0,1)$. We then evaluate the trained transformer under two representative distributional shifts:

1. The measurement matrix $H_i$ is sampled from a uniform distribution $\mathcal{U}[0,\ 3]$;

2. The state transition matrix $F$ is generated using Strategy 2 instead of Strategy 1.

As shown in Fig. 9, the transformer maintains low MSPD in both settings, indicating robustness of in-context learning to shifts in both measurement geometry and underlying dynamics.

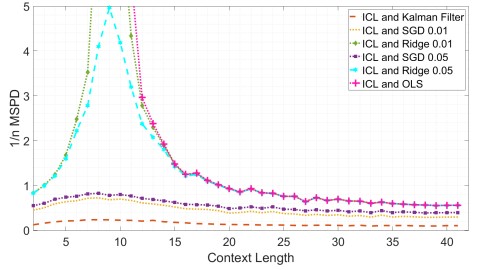 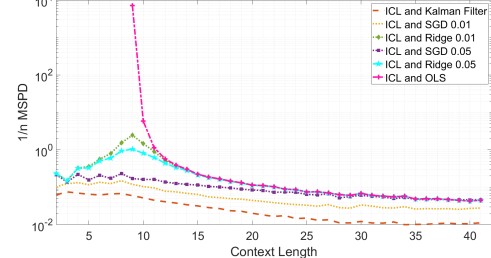

(a) MSPD under uniform sampling of $H_i \sim \mathcal{U}(0,\ 3)$.     (b) MSPD under shift from Strategy 1 to Strategy 2 for generating $F$.

Figure 9: *Evaluation of a transformer trained on Gaussian-distributed systems under representative out-of-distribution shifts.*

## E.2    Transformer performance across varying state dimensions

We next assess how transformer performance varies with the dimensionality of the latent state. The transformer is trained using default settings and again with $F$ sampled via Strategy 1. During evaluation, we fix the context length to 40 and vary the state dimension from 2 to 8.

Figure 10 shows the MSPD (normalized by state dimension) between the transformer and various baseline methods. We find that:

- The gap between the transformer and Kalman filter remains nearly constant across dimensions.

- The discrepancy between the transformer and SGD / Ridge regression grows with increasing state dimension, reflecting the degraded performance of those baselines.

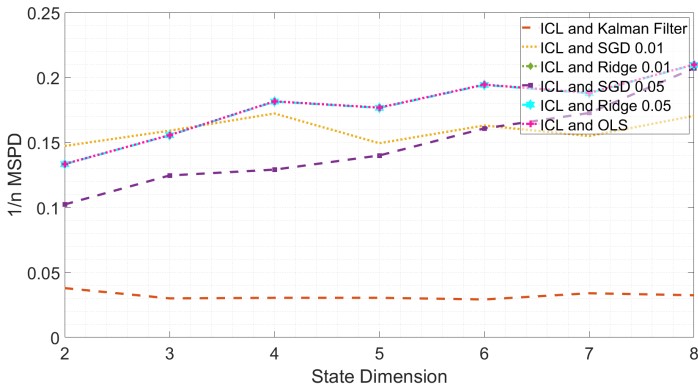

Figure 10: *Normalized MSPD between transformer and various baselines as a function of state dimension. Context length is fixed to 40.*

### E.3 Justification of the choice of the learning rates for SGD

To identify the optimal learning rate $\alpha$ for stochastic gradient descent (SGD) and the regularization parameter $\lambda$ for ridge regression, we performed a sweep over a logarithmic range from $10^{-5}$ to $10^0$. Based on mean-squared prediction difference (MSPD) between each baseline and the transformer, we found that $\alpha = 0.01$ and $\lambda = 0.01$ consistently yielded the best performance across most context lengths, with only occasional exceptions. For consistency and clarity, we fixed both parameters to 0.01 in all experiments reported in the main paper.

### E.4 Multi-step Prediction

In this section, we evaluate whether a transformer trained solely for one-step prediction can generalize to longer horizons via recursive in-context inference. Specifically, we consider the task of predicting the future output $y_{N-1+\tau} = h_{N-1}x_{N-1+\tau}$, given a context of $N-1$ input–output pairs and system parameters. The initial context is structured as

$$\begin{bmatrix} 0 & 0 & \sigma^2 & 0 & y_1 & 0 & y_2 & ... & 0 & y_{N-1} \\ F & Q & 0 & h_1^T & 0 & h_2^T & 0 & ... & h_{N-1}^T & 0 \end{bmatrix}. \tag{51}$$

To perform $\tau$-step prediction, we recursively apply the transformer $\tau$ times. At each step $i$, the predicted output $\hat{y}_i$ is appended to the input stream. To isolate the prediction challenge, we freeze the final measurement matrix $h_{N-1}$ and reuse it for all future steps $t > N - 1$. For instance, to generate $\hat{y}_{N-1+\tau}$, we feed the transformer the extended input

$$\begin{bmatrix} 0 & 0 & \sigma^2 & 0 & y_1 & 0 & y_2 & \cdots & y_{N-1} & 0 & \hat{y}_N & \cdots & \hat{y}_{N-1+\tau-1} & 0 \\ F & Q & 0 & h_1^\top & 0 & h_2^\top & 0 & \cdots & 0 & h_{N-1}^\top & 0 & \cdots & 0 & h_{N-1}^\top \end{bmatrix}. \tag{52}$$

We adopt a similar setup for baseline methods: stochastic gradient descent (SGD) and ridge regression are trained with input–output pairs where the output label corresponds to a measurement $\tau$ steps ahead. Experiments are conducted under Strategy 1 for system generation, using three values of process noise variance: $\sigma_q^2 \in 0.025, 0.1, 0.5$. These control the entries of the process noise covariance matrix $Q$.

Figure 11 summarizes the results. As expected, prediction error increases with both horizon length $\tau$ and noise level $\sigma_q^2$. Nevertheless, the transformer maintains performance closely aligned with the Kalman filter and significantly outperforms regression-based methods, even in challenging multi-step prediction settings.

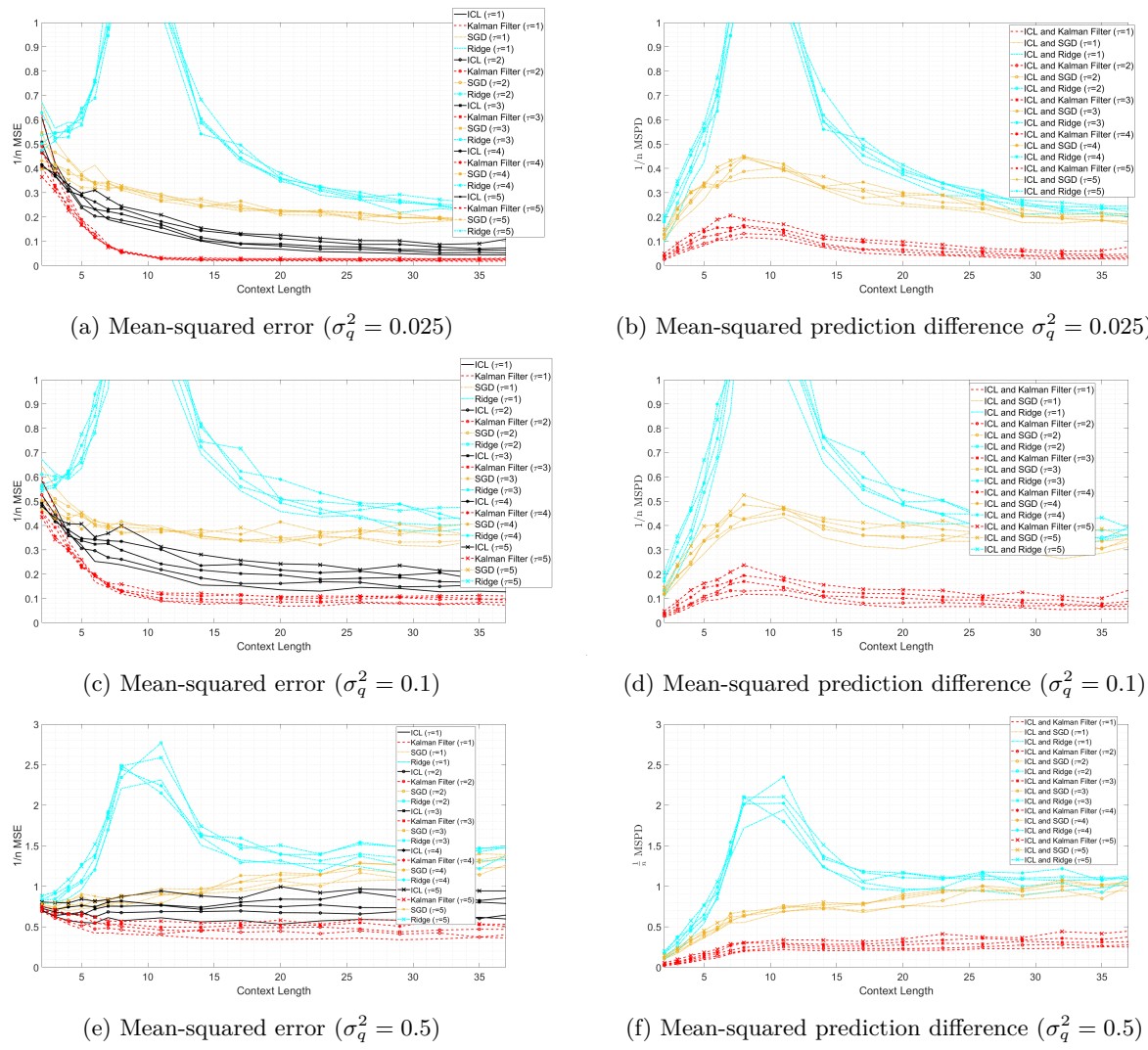

Figure 11: *Multi-step prediction performance on linear dynamical systems (Strategy 1). For each value of process noise variance $\sigma_q^2$, we evaluate MSE (left) and MSPD (right) between transformer predictions and baselines over horizons $\tau = 1$ to 5. The transformer (ICL) tracks the Kalman filter closely and outperforms Ridge regression and SGD, demonstrating its ability to propagate uncertainty over multiple steps.*

### E.5 Is the transformer utilizing the parameters of the state-space model?

In the main text, we investigated the transformer's performance on linear systems when major components of the state-space model (specifically, $F$, $Q$, and $R$) were withheld from the context. Here, we take a step further and evaluate the case where *all* parameters of the state-space model, including the measurement matrices $H_t$, are excluded. That is, the transformer must predict $\hat{y}_N$ given only the past measurements $\hat{y}_1, \hat{y}_2, \ldots, \hat{y}_{N-1}$ as context. As shown in Fig. 12, in this setting the performance deteriorates sharply, as indicated by the significant increase in the resulting MSE.

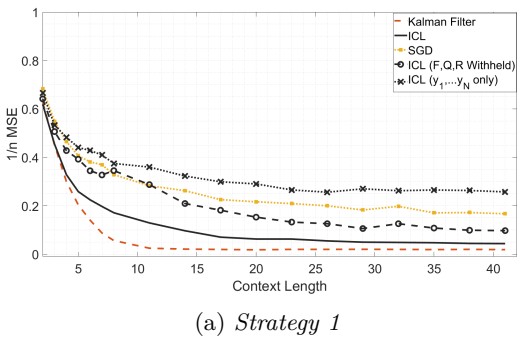 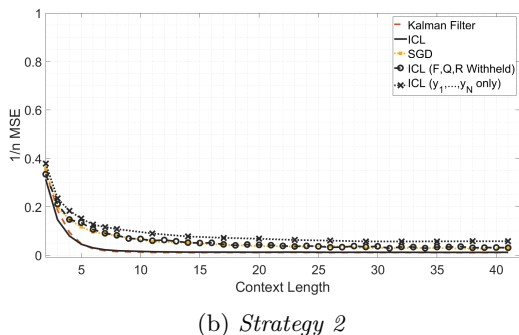

(a) *Strategy 1*                    (b) *Strategy 2*

Figure 12: *Performance of in-context learning for a system where the state dimension is 8, under two system generation strategies. As key parameters of the state-space model are omitted from the context, the transformer's performance degrades steadily. The gap between ICL and the Kalman filter is more pronounced under Strategy 1, where inference is more challenging.*

## F    Extended Results and Analysis for Nonlinear Systems

This section provides additional details for the nonlinear systems analyzed in Section 4.3. Specifically, we consider systems governed by nonlinear state-space dynamics of the form:

$$x_{t+1} = f_\eta(x_t) + q_t \tag{53}$$
$$y_t = H_t x_t + r_t, \tag{54}$$

where $f_\eta(\cdot)$ is a nonlinear transition function parameterized by $\eta = [\eta_1, \ldots, \eta_w]$, and $q_t$, $r_t$ denote the process and observation noise, respectively.

A classical approach to estimating the latent states in such systems is by means of the Extended Kalman Filter (EKF), which linearizes the dynamics around the current estimate at each time step. The EKF proceeds via the following prediction and update recursions:

**Prediction Step:**

$$\hat{x}_t^- = f_\eta(\hat{x}_{t-1}^+) \tag{55}$$
$$\hat{P}_t^- = \tilde{F}_x \hat{P}_{t-1}^+ \tilde{F}_x^\top + Q \tag{56}$$

**Update Step:**

$$K_t = \hat{P}_t^- H_t^\top (H_t \hat{P}_t^- H_t^\top + R)^{-1} \tag{57}$$
$$\hat{x}_t^+ = \hat{x}_t^- + K_t(y_t - H_t \hat{x}_t^-) \tag{58}$$
$$\hat{P}_t^+ = (I - K_t H_t)\hat{P}_t^- \tag{59}$$

Here, $\tilde{F}_x$ denotes the Jacobian of $f_\eta$ evaluated at $\hat{x}_{t-1}^+$.

### F.1    System 1: Nonlinear State Transitions with tanh Dynamics

We consider a specific nonlinear state transition function $\mathbf{f}_\eta(\mathbf{x}) = [\eta_1 \tanh(\eta_2 \mathbf{x_1}), \ldots, \eta_1 \tanh(\eta_2 \mathbf{x_n})]$, where the parameters $\eta_1$ and $\eta_2$ are independently sampled from the uniform distribution $\mathcal{U}[-1, 1]$. For this function, the Jacobian with respect to the state vector is diagonal and given by

$$\tilde{F}_x = \eta_1 \eta_2 \cdot \text{diag}\left(1 - \tanh^2(\eta_2 x_1), \ldots, 1 - \tanh^2(\eta_2 x_n)\right).$$

This structure enables efficient computation of the EKF update steps using element-wise operations. We argue that a transformer can emulate these operations in-context, provided that the input sequence is formatted as

$$\begin{bmatrix} \eta_1 & \eta_2 & 0 & \sigma^2 & 0 & y_1 & \ldots & y_{N-1} & 0 \\ 0 & 0 & Q & 0 & h_1^T & 0 & \ldots & 0 & h_N^T \end{bmatrix}. \tag{60}$$

Prior work (e.g., Akyürek et al. (2023)) has observed that the GeLU nonlinearity can approximate scalar multiplication and, with sufficient additive bias, act nearly as an identity function. Building on this observation, we construct an expression that closely approximates $\tanh(x)$ using GeLU activations as

$$\frac{\sqrt{\frac{\pi}{2}}x}{2} \tanh(x + cx^3) \approx \text{GeLU}\left(\sqrt{\frac{\pi}{2}}x\right) - \text{GeLU}\left(\frac{\sqrt{\frac{\pi}{2}}x}{2} + N_b\right) + N_b,$$

where $N_b \gg 1$ is a large bias term and $c = \frac{\pi}{2} \cdot 0.044715$. This identity leverages the approximation of $\tanh(\cdot)$ as a difference of GeLU activations with shifted inputs.

Since transformers can effectively bypass softmax via large biases (making attention weights nearly one-hot), this approximation can be implemented by a single attention head. In the regime we study, $x + cx^3 \approx x$, so the net operation approximates $\tanh(x)$ directly. When combined with the transformer's ability to realize primitive arithmetic operations such as **Mul()**, **Div()**, and **Aff()**, this suffices to emulate the nonlinear update steps of the Extended Kalman Filter for the system under consideration.

### F.2 System 2: Target Tracking with Unknown Turn Rate

We evaluate the transformer's in-context learning capabilities on a practical nonlinear system: maneuvering target tracking with an unknown turning rate Piché et al. (2012). The state vector is $x_t = [a_t, \dot{a}_t, b_t, \dot{b}_t, \omega_t]$, where $(a_t, b_t)$ is the position, $(\dot{a}_t, \dot{b}_t)$ is the velocity, and $\omega_t$ is the angular turn rate.

The state transition model is

$$x_{t+1} = F_t x_t + q_t, \tag{61}$$

where $F_t$ is a nonlinear function of $\omega_t$

$$F_t = \begin{bmatrix} 1 & \frac{\sin(\omega_t \Delta_t)}{\omega_t} & 0 & \frac{\cos(\omega_t \Delta_t)-1}{\omega_t} & 0 \\ 0 & \cos(\omega_t \Delta_t) & 0 & -\sin(\omega_t \Delta_t) & 0 \\ 0 & \frac{1-\cos(\omega_t \Delta_t)}{\omega_t} & 1 & \frac{\sin(\omega_t \Delta_t)}{\omega_t} & 0 \\ 0 & \sin(\omega_t \Delta_t) & 0 & \cos(\omega_t \Delta_t) & 0 \\ 0 & 0 & 0 & 0 & 1 \end{bmatrix}$$

and the process noise covariance is

$$Q = \begin{bmatrix} q_1 M & 0 & 0 \\ 0 & q_1 M & 0 \\ 0 & 0 & q_2 \end{bmatrix}, \quad M = \begin{bmatrix} \Delta_t^3/3 & \Delta_t^2/2 \\ \Delta_t^2/2 & \Delta_t \end{bmatrix}.$$

The measurement vector consists of noisy versions of the polar coordinates of the target, i.e.,

$$y_t = \begin{bmatrix} \sqrt{a_t^2 + b_t^2} \\ \arctan(\frac{b_t}{a_t}) \end{bmatrix} + r_t.$$

We set $\Delta_t = 0.1$ and draw each simulation's parameters as: $q_1 \sim \mathcal{U}[0, 0.1]$, $q_2 \sim \mathcal{U}[0, 0.00025]$, $\sigma_1^2 \sim \mathcal{U}[0, 0.025]$, $\sigma_2^2 \sim \mathcal{U}[0, 0.000016]$, and the initial state from $x_0 \sim \mathcal{N}([0, 10, 0, -5, -0.053]^T, Q)$. The transformer is trained to predict $y_N$ from an input formatted as

$$\begin{bmatrix} 0 & \sigma^2 & y_1 & y_2 & \ldots & y_{N-1} & \\ Q & 0 & 0 & 0 & \ldots & 0 & \end{bmatrix}. \tag{62}$$

To implement the Extended Kalman Filter, we compute the Jacobians as

$$
F_x = \begin{bmatrix}
1 & \frac{\sin(\omega_t \Delta_t)}{\omega_t} & 0 & \frac{\cos(\omega_t \Delta_t)-1}{\omega_t} & \dot{a}_t f_{1t} + \dot{b}_t f_{2t} \\
0 & \cos(\omega_t \Delta_t) & 0 & -\sin(\omega_t \Delta_t) & -\Delta_t sin(\omega_t \Delta_t)\dot{a}_t - \Delta_t cos(\omega_t \Delta_t)\dot{b}_t \\
0 & \frac{1-\cos(\omega_t \Delta_t)}{\omega_t} & 1 & \frac{\sin(\omega_t \Delta_t)}{\omega_t} & -\dot{a}_t f_{2t} + \dot{b}_t f_{1t} \\
0 & \sin(\omega_t \Delta_t) & 0 & \cos(\omega_t \Delta_t) & \Delta_t cos(\omega_t)\dot{a}_t - \Delta_t sin(\omega_t \Delta_t)\dot{b}_t \\
0 & 0 & 0 & 0 & 1
\end{bmatrix}
$$

and

$$
H_x = \begin{bmatrix}
\frac{a_t}{\sqrt{a_t^2+b_t^2}} & 0 & \frac{b_t}{\sqrt{a_t^2+b_t^2}} & 0 & 0 \\
-\frac{b_t}{a_t^2+b_t^2} & 0 & -\frac{a_t}{a_t^2+b_t^2} & 0 & 0
\end{bmatrix},
$$

where

$$
f_{1k} = \frac{\omega_t \Delta_t cos(\omega_t \Delta_t) - sin(\omega_t \Delta_t)}{\omega_t^2} \text{ and } f_{2k} = \frac{1 - \omega_k \Delta_t sin(\omega_t \Delta_t) - cos(\omega_t \Delta_t)}{\omega_t^2}.
$$

### F.3 RAW Operator Approximation of Smooth Nonlinearities

The set of operations implementable by the RAW operator, including multiplication, affine combinations, and repeated composition, is sufficient to construct any polynomial. Indeed, monomials of the form $cx^k$ can be computed via one scalar multiplication followed by $k-1$ applications of the multiplication operator. Once computed, polynomial terms can be linearly combined using the affine operator. As a result, functions expressible via Taylor expansions can be approximated by transformers to arbitrary accuracy. For example:

$$
\sin(x) \approx x - \frac{x^3}{3!} + \frac{x^5}{5!} - \frac{x^7}{7!} + \frac{x^9}{9!} - \frac{x^{11}}{11!} + \cdots \tag{63}
$$

$$
\cos(x) \approx 1 - \frac{x^2}{2!} + \frac{x^4}{4!} - \frac{x^6}{6!} + \frac{x^8}{8!} - \frac{x^{10}}{10!} + \frac{x^{12}}{12!} + \cdots \tag{64}
$$

$$
\tanh(x) \approx x - \frac{x^3}{3} + \frac{2x^5}{15} - \frac{17x^7}{315} + \frac{62x^9}{2835} - \frac{1382x^{11}}{155925} + \cdots \tag{65}
$$

Similar series exist for $arctan(x)$ and $\sqrt{x}$, which also appear in the nonlinear systems considered in this work. Since transformers can represent these polynomial approximations with sufficient depth and width, this supports the claim that they are capable of learning to approximate the behavior of extended Kalman filtering in nonlinear dynamical systems.

### F.4 Additional Non-linear State Transitions

Figure 13 presents results for several variations of nonlinear dynamical systems, each differing in the form of the state transition function:

1. **System I:** $f_\eta(x) = F \sin(2x) + q_t$, with $n = 2$ and $m = 2$. System parameters $F$, $Q$, and $R$ are identical to those used in System 1 of the main paper.

2. **System II:** $f_\eta(x) = F\sigma(2x) + q_t$, with $n = 2$ and $m = 2$, where $\sigma(\cdot)$ denotes the sigmoid function. All other system parameters match those of System 1.

3. **System III:** $f_\eta(x) = F \tanh(2x) + q_t$, with $n = 8$ and $m = 1$. System parameters remain consistent with System 1.

4. **System IV:** $f_\eta(x) = F \tanh(2x) + \frac{2}{9} \exp(-x^2) + q_t$, with $n = 2$ and $m = 2$, and system parameters again matching those of System 1.

In all cases, the entries of the measurement matrix $H_t$ are drawn from an isotropic Gaussian distribution. These experiments confirm that the transformer's performance generalizes across a range of nonlinearities. In all four cases, the MSE closely matches or outperforms the Extended Kalman Filter and tracks the performance of the particle filter, particularly as context length increases. This consistency highlights the transformer's capacity to adapt its inference strategy to diverse dynamical structures.

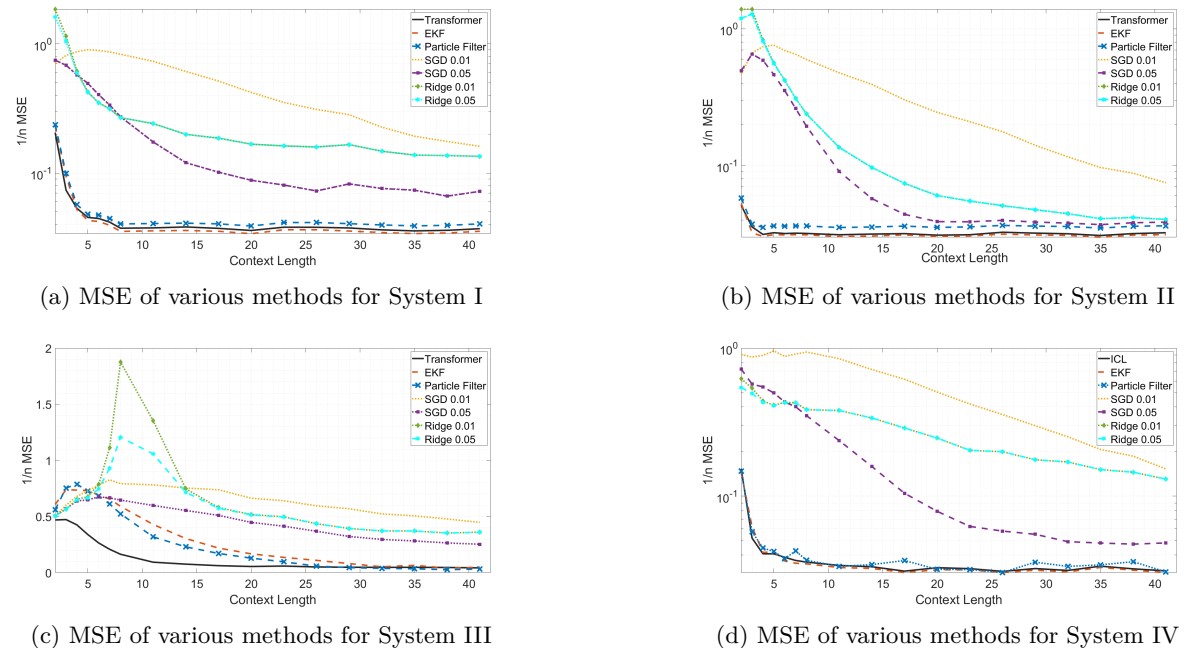

(a) MSE of various methods for System I

(b) MSE of various methods for System II

(c) MSE of various methods for System III

(d) MSE of various methods for System IV

Figure 13: *Transformer performance on nonlinear systems with varying state transition functions. Evaluation is conducted for each of the systems described above, illustrating that in-context learning remains effective across a range of nonlinearities.*

## G  Derivation of the RAW Operator

To make this work self-contained, we briefly summarize how the RAW operator can be constructed and used to emulate key computational primitives within transformer architectures. This derivation follows the treatment in Appendix C of Akyürek et al. (2023), to which we refer interested readers for additional technical details and formal proofs.

### G.1  Embedding Layer Design and Positional Control

We assume the existence of a linear embedding layer immediately preceding the transformer layers. This embedding layer can be parameterized to include extra scratch space necessary for intermediate computations. Specifically, we use the embedding matrix

$$W_e = \begin{bmatrix} I_{(d+1)\times(d+1)} & \mathbf{0} \\ \mathbf{0} & \mathbf{0} \end{bmatrix}, \tag{66}$$

which embeds the input into the first $d + 1$ dimensions and reserves the remaining entries for auxiliary operations.

Following Akyürek et al. (2023), attention is directed using position embeddings that encode keys and queries. These are structured as

$$p_i = \begin{bmatrix} \mathbf{0}_{d+1} & k_i^0 & q_i^0 & \dots & k_i^{(L)} & q_i^{(L)} & \mathbf{0}_{H-2pT-1} \end{bmatrix},$$

where each $k_i^l, q_i^l \in \mathbb{R}^p$ controls the attention for head $l$ at time step $i$. This setup allows queries and keys to be extracted via

$$W_Q^l = \begin{bmatrix} \mathbf{0} & \ldots & \mathbf{0} & I_{p \times p} & \mathbf{0} & \ldots \end{bmatrix}, \quad W_K^l = \begin{bmatrix} \mathbf{0} & \ldots & \mathbf{0} & I_{p \times p} & \mathbf{0} & \ldots \end{bmatrix},$$

where $I_{p \times p}$ selects the appropriate query or key block.

Using this formulation, two common attention patterns can be encoded:

1. **Attend to the previous token.** To enforce attention from time step $i$ to $i-1$, set

$$k_i = e_i \tag{67}$$
$$q_i = Ne_{i-1}, \tag{68}$$

   where $e_i$ is the $i^{\text{th}}$ standard basis vector and $N \gg 1$ ensures the softmax selects the intended token.

2. **Attend to a fixed token.** To force every future token to attend to token $t$, define

$$K(i) = \begin{cases} \{t\}, & i > t \\ \emptyset, & \text{otherwise} \end{cases} \tag{69}$$

   and set the positional encodings to

$$k_i = \begin{cases} -N, & i \neq t \\ N, & i = t \end{cases}, \quad q_i = N. \tag{70}$$

## G.2   Utilizing Nonlinearities

The nonlinear components of transformer layers can be exploited to implement element-wise arithmetic operations with high accuracy. Below, we restate three key results that underpin the construction of such operations within transformer blocks.

1. **Element-wise multiplication via GeLU.** The GeLU activation can approximate multiplication,

$$\sqrt{\frac{\pi}{2}} \left( \text{GeLU}(x+y) - \text{GeLU}(x) - \text{GeLU}(y) \right) = xy + \mathcal{O}(x^3 + y^3), \tag{71}$$

   enabling scalar products to be implemented through combinations of GeLU activations and additive operations.

2. **Bypassing GeLU.** The GeLU nonlinearity can be made approximately linear by shifting its input by a large constant according to

$$\text{GeLU}(N+x) - N \approx x, \qquad N \gg 1. \tag{72}$$

   This allows a transformer head to effectively implement identity transformations when needed.

3. **Bypassing layer normalization.** With appropriate padding and large additive constants, layer normalization can be suppressed, i.e.,

$$\sqrt{\frac{2}{L}} N \cdot \lambda([\mathbf{x}, \ N, \ -N - \sum \mathbf{x}, \ \mathbf{0}]) \approx [\mathbf{x}, \ 2N, \ -2N - \sum \mathbf{x}, \ \mathbf{0}], \tag{73}$$

   where $\lambda(\cdot)$ denotes layer normalization and $\mathbf{x} \in \mathbb{R}^L$. This result shows that layer norm can be effectively neutralized by padding the input with tailored constants.

### G.3 Parameterizing the RAW Operator

Recall the transformer layer formulation:

$$b_\gamma^{(l)} = \text{Softmax}\left((W_\gamma^Q G^{(l-1)})^\top (W_\gamma^K G^{(l-1)})\right)(W_\gamma^V G^{(l-1)}), \tag{74}$$

$$A^{(l)} = W^F[b_1^{(l)}, b_2^{(l)}, \ldots, b_B^{(l)}], \tag{75}$$

$$G^{(l)} = W_1 \sigma\left(W_2 \lambda\left(A^{(l)} + G^{(l-1)}\right)\right) + A^{(l)} + G^{(l-1)}. \tag{76}$$

We aim to show that a transformer layer with parameters $\theta = \{W^F, W_1, W_2, (W^Q, W^K, W^V)_m\}$ can approximate the RAW operator. Following the construction in Akyürek et al. (2023), this can be achieved using just two attention heads. Key and query matrices are set to implement the token access pattern $K(i)$, enabling the approximation of

$$\frac{W_a}{|K(i)|} \sum_{k \in K(i)} G_k^{(l)}[r]. \tag{77}$$

The first head is configured as follows:

- $W_1^Q, W_1^K$ encode the attention pattern $K(i)$.

- $W_1^V$ is sparse, with nonzero entries $(W_1^V)_{t[m], \ r[n]} = (W_a)_{m,n}$, $m = 1, \ldots, |t|$, $n = 1, \ldots, |r|$.

- This ensures that only entries in $t$ are modified, while others remain unchanged, i.e.,

$$(A_i^{(l)} + G_i^{(l)})_t = \frac{W_a}{|K(i)|} \sum_{k \in K(i)} G_k^{(l)}[r], \quad (A_i^{(l)} + G_i^{(l)})_{t' \notin t} = G_i^{(l)}[t' \notin t]. \tag{78}$$

To cancel the residual in $t$, the second head is set up as

$$(W_2^Q, \ W_2^K) : K(i) = i, \quad (W_2^V)_{t[m], \ r[n]} = -1. \tag{79}$$

The feedforward projection combines the two heads as

$$(W^F)_{t[m], \ t[m]} = 1, \quad (W^F)_{t[m], \ t[m]+H} = -1. \tag{80}$$

This yields the attention output matching expression (77), isolating changes to $t$.

The complete RAW operator is defined as

$$G_{i,w}^{(l+1)} = W_o\left(\left[\frac{W_a}{|K(i)|} \sum_{k \in K(i)} G_k^{(l)}[r]\right] \circ W G_i^{(l)}[s]\right), \tag{81}$$

$$G_{i,j \notin w}^{(l+1)} = G_{i,j \notin w}^{(l)}. \tag{82}$$

Let $m_i$ denote the output of the second MLP layer. To emulate the RAW operator, we require

$$(m_i)_{t' \in w} = W_o\left(\left(\frac{W_a}{|K(i)|} \sum_{k \in K(i)} G_k^{(l)}[r]\right) \circ W G_i^{(l)}[s]\right) - G_i^{(l)}[w] - A_i[w], \tag{83}$$

$$(m_i)_{t' \in t} = -G_i^{(l)}[t] - A_i[t], \tag{84}$$

$$(m_i)_{t' \notin t \cup w} = 0. \tag{85}$$

The exact structure of the MLP depends on whether $\circ$ represents addition or element-wise multiplication.

### G.3.1   Additive Operator ($\circ = +$)

Let $u_i$ be the output of the first MLP layer. Since the GeLU non-linearity can be bypassed by adding a large bias term $N$, we aim to configure $u_i$ such that

$$(u_i)_{\hat{t}} = W G_i^{(l-1)}[s] + G_i^{(l)}[t] + A_i[t] + N, \tag{86}$$

$$(u_i)_t = -(G_i^{(l)}[t] + A_i[t]) + N, \tag{87}$$

$$(u_i)_w = -(G_i^{(l)}[w] + A_i[w]) + N, \tag{88}$$

$$(u_i)_{j \notin (t \cup \hat{t} \cup w)} = -N. \tag{89}$$

This is achieved by setting the first MLP layer's weights $W_1$ as

$$(W_1)_{\hat{t}[m],\ s[n]} = W_{m,n}, \tag{90}$$

$$(W_1)_{\hat{t}[m],\ t[n]} = 1, \tag{91}$$

$$(W_1)_{t[m],\ t[m]} = -1, \tag{92}$$

$$(W_1)_{w[m],\ w[m]} = -1, \tag{93}$$

and its bias $b_1$ as

$$(b_1)_{\hat{t}},\ (b_1)_t,\ (b_1)_w = N, \tag{94}$$

$$(b_1)_{j \notin (t \cup \hat{t} \cup w)} = -N. \tag{95}$$

The second MLP layer is configured to combine and cancel terms as needed,

$$(W_2)_{w[m],\ \hat{t}[n]} = (W_o)_{m,n}, \tag{96}$$

$$(W_2)_{t[m],\ t[m]} = 1, \quad (W_2)_{w[m],\ w[m]} = 1, \tag{97}$$

with bias

$$(b_2)_{w[m]} = -N \sum_j (W_o)_{m,j} - N, \tag{98}$$

$$(b_2)_{t[m]} = -N, \tag{99}$$

$$(b_2)_{j \notin t} = 0. \tag{100}$$

### G.3.2   Multiplicative Operator ($\circ = *$)

To approximate multiplication, we introduce three auxiliary hidden slots $t_a, t_b, t_c$. The first MLP layer's output is configured as

$$(u_i)_{t_a} = \frac{W G_i^{(l)}[s] + G_i^{(l)}[t] + A_i[t]}{N}, \tag{101}$$

$$(u_i)_{t_b} = \frac{G_i^{(l)}[t] + A_i[t]}{N}, \tag{102}$$

$$(u_i)_{t_c} = \frac{W(G_i^{(l)}[s] + A_i[s])}{N}, \tag{103}$$

$$(u_i)_t = -(G_i^{(l)}[t] + A_i[t]) + N, \tag{104}$$

$$(u_i)_w = -(G_i^{(l)}[w] + A_i[w]) + N, \tag{105}$$

$$(u_i)_{j \notin (t \cup t_a \cup t_b \cup t_c \cup w)} = -N. \tag{106}$$

This behavior is achieved by setting the first MLP layer's parameters as

$$(W_1)_{t_a[m],\ s[n]} = \frac{W_{m,n}}{N}, \quad (W_1)_{t_a[m],\ t[n]} = \frac{1}{N}, \tag{107}$$

$$(W_1)_{t_b[m],\ t[m]} = \frac{1}{N}, \quad (W_1)_{t_c[m],\ s[m]} = \frac{1}{N}, \tag{108}$$

$$(W_1)_{t[m],\ t[m]} = -1, \quad (W_1)_{w[m],\ w[m]} = -1, \tag{109}$$

and bias vector

$$(b_1)_{t \cup t_a \cup t_b \cup t_c} = 0, \quad (b_1)_{t \cup w} = N, \quad (b_1)_{j \notin (t \cup t_a \cup t_b \cup t_c \cup w)} = -N. \tag{110}$$

The second MLP layer computes the product via

$$(W_2)_{w[m],\ t_a[n]} = (W_o)_{m,n} N^2 \sqrt{\frac{\pi}{2}}, \tag{111}$$

$$(W_2)_{w[m],\ t_b[n]} = -(W_o)_{m,n} N^2 \sqrt{\frac{\pi}{2}}, \tag{112}$$

$$(W_2)_{w[m],\ t_c[n]} = -(W_o)_{m,n} N^2 \sqrt{\frac{\pi}{2}}, \tag{113}$$

$$(W_2)_{w[m],\ w[m]} = 1, \quad (W_2)_{t[m],\ t[m]} = 1, \tag{114}$$

with bias

$$(b_2)_{t \cup w} = N, \quad (b_2)_{j \notin (t \cup w)} = 0. \tag{115}$$

**Division via LayerNorm.** To compute $\mathbf{y}/c$ from a structured input vector $[c, \mathbf{y}, \mathbf{0}]^T$, one can use the approximation

$$\sqrt{\frac{2}{L}} MN\ \lambda\left([Nc,\ \mathbf{y}/M,\ -Nc - \sum \mathbf{y}/M,\ \mathbf{0}]\right) \approx [MN,\ \mathbf{y}/c,\ -MN - \mathbf{y}/c,\ \mathbf{0}], \tag{116}$$

where $M, N \gg 1$. The desired quotient $\mathbf{y}/c$ can then be isolated through appropriate weight selection in the feedforward layer.

### G.4 Expressing Core Operations via the RAW Operator

Three fundamental operations used throughout this paper can be expressed as special cases of the `RAW` operator. Specifically:

$$\begin{aligned}
\mathbf{dot}(G; (i,j), (i',j'), (i'',j'')) &= \mathbf{Mul}(G; 1, |i-j|, 1, (i,j), (i',j'), (i'', i''+1)) \\
&= \mathtt{RAW}(G;\ *,\ W = I,\ W_a = I,\ W_o = \mathbb{1}^T,\ s = (i,j),\ r = (i',j'), \\
&\qquad w = (i'', i''+1),\ K = \{(t, \{t\})\ \forall t\}) \tag{117} \\
\mathbf{Aff}(G; (i,j), (i',j'), (i'',j''), W_1, W_2, b) &= \mathtt{RAW}(G;\ +,\ W = W_1,\ W_a = W_2,\ W_o = I,\ b_0 = b, \\
&\qquad s = (i,j),\ r = (i',j'),\ w = (i'', i''+1),\ K = \{(t, \{t\})\ \forall t\}) \tag{118} \\
\mathbf{mov}(G;\ s,\ t,\ (i,j), (i',j')) &= \mathtt{RAW}(G;\ +,\ W = 0,\ W_a = I,\ W_o = I, \\
&\qquad s = (),\ r = (i',j'),\ w = (i,j),\ K = \{(t, \{s\})\}) \tag{119}
\end{aligned}$$

These equivalences follow directly by substitution. Moreover, the **dot** operator naturally extends to parallel execution, enabling efficient matrix-level implementation of **Mul**.

