# OpenReview forum: "Transformers as Implicit State Estimators: In-Context Learning in Dynamical Systems"
_TMLR — Accepted by TMLR_

### Review · Reviewer_jnXo · 2025-09-26

**Summary Of Contributions:**

The authors study the ability of transformers to learn dynamical systems using in-context learning (ICL). Given a set of past input-output pairs generated by a linear or possibly non-linear dynamical system, the objective is to predict the current output given the current input. For a linear dynamical system (LDS) with Gaussian inputs, the Kalman filter (KF) is the optimal estimator, while approximate methods such as the extended Kalman filter (EKF) or particle filter are typically used in practice when dealing with non-linear dynamics.

From my perspective, I see two main contributions in this paper:
1. The authors reformulate the Kalman filter using operations implementable by a transformer, demonstrating that a transformer could potentially learn to emulate a KF. They then provide experimental results demonstrating that, indeed, transformers can indeed do this.
2. The authors demonstrate that transformers can in-context learn to predict outputs in non-linear dynamical systems with accuracy competitive with commonly-used approximate methods such as the EKF and particle filtering.

## Strengths
- The authors provide a combination of theoretical and empirical contributions demonstrating that a transformer can learn to emulate a KF using ICL.
- The paper is well written and easy to understand, even for a reader who may not be familiar with the latest research on transformers and ICL. The important elements of the Read-Arithmetic-Write (RAW) framework (Akyürek et al., 2023) used to establish the theoretical results are summarized in an appendix.

## Weaknesses
- The plots are trying to show too many things. This is especially true for Figure 5, where there are additional curves for $n=8$.
- The meaning of the context length for the KF and variants is unclear.
- The authors consider only one-step forward prediction.

**Additional Comments:**

Suggested minor revisions:
- Line under equation (20): Please add that you are indexing your tokens beginning with 0, not 1. If using indexing beginning with 1, then the $(2n+1)$th token is $\sigma^2$, not $y_1$.
- Figure 6(b): ICL and Kalman filter should probably be ICL and EKF.

**Audience:**

Yes

**Audience Explanation:**

Yes, a large portion of the machine learning community is currently interested in understanding what transformers can and can't learn, and this paper contributes to that understanding.

**Claims And Evidence:**

Yes

**Claims Explanation:**

Yes, with both theoretical and empirical evidence.

**Requested Changes:**

- Is there a need to have both SGD and Ridge with two learning rates along with OLS in the plots? The OLS behavior doesn't seem too different from Ridge, so I suggest including only SGD and Ridge with a single learning rate, and if desired, including all of the curves in additional plots in the appendix.
- The x axis on all of the figures shows the context length, for which the meaning is clear for ICL, but not for the Kalman filter and variants. Please explain the meaning of context length for the Kalman filter, EKF, and particle filter.
- The MSE for ICL and MSPD between ICL and the Kalman filter seems to hit a floor beyond context length of around 30 and does not continue to decrease. Do you have some sense for what is happening and whether the error is due to bias or variance in the ICL's predictions?
- Figure 7 caption: Why do you state that the transformer performance matches the particle filter? It looks to be doing much better than the particle filter.
- Please discuss whether it is possible to perform prediction beyond one time step. For example, if $H_t$ remains fixed over time, this seems like it should be possible.

---

### Review · Reviewer_qRSw · 2025-12-15

**Summary Of Contributions:**

The paper explicitly studies whether transformers can be trained to emulate time series filter algorithms by feeding the state-space system to the transformer as inputs. The paper shows that transformers can fit non-linear dynamics well, but struggles a bit with linear dynamics.

**Additional Comments:**

The notation in the paper could be made more consistent. In sec 2 the layer inputs are first described as G^l, then x_t, and finally by q^l [which in appendix is again G^l]. Since the paper is proposing a unified treatment of transformers and state-space systems, it would make sense to use a unified notation as well.

How is the kalman filter of eqs 13-17 different from eqs 22-27? Why do we need both?

Algorithm 1 is incomprehensible to me. This should be annotated to make sense of this. If alg 1 is supposed to be used as just a code listing, it would be better given as eg. python code.

Moreover, I'm not totally sure how a transformer is actually used in this paper. I don't understand how the mul/div/aff/trans operations now implement a transformer that executes eqs 22-27 or eq 28. I don't see how eq 28 goes into alg1. Do we still use standard transformers in this paper, or is the transformer some special architecture? What kind?

The network used seems quite large, and the batch size as well. Was it necessary to have such larger transformers and batches to make this work? If transformers in some sense naturally align with kalman filters, shouldn't we expect much simpler networks to work as well?

Why do Ridge and OLS fare so poorly in fig1? I would assume that these are gold standard estimators, and should give good performance. Can you explain?

It seems that the transformer can't predict as well as kalman filter in Fig1. Why not? I thought that we expected transformer to be theoretically capable of solving this problem, and since we explicitly train a transformer to accomplish this task, it should do quite well in practise as well.

Overall I'm a bit confused of this paper's setting. The paper assumes a Kalman algorithm, and feeds all the components of it to a transformer as input, and tries to fit the state-space system. This is an unusual way of using a transformer, so what does this precisely tell us? I was under the impression that the purpose of this paper is to show that transformers use similar mechanisms as kalman filters, and are thus theoretically equally capable. I also thought the paper was about showing how a standard GPT2 emulates kalman filters (without knowing that it's supposed to be doing it).

Finally, even if the transformer is able to fit the state space using the kalman algorithm as input, how do we know the transformer is even using the input? What if the transformer is solving the system in it's own "style" and disregards, corrupts or abuses some of the in-context input?

The paper claims that ridge regression suffers from double descent. I'm not sure if I agree. Doesn't the regulariser mitigate the noise fitting problem of double descent?

Why the context length is limited to 40? Why not go much further, eg 1000?

Intuitively it would make sense to compare to a vanilla transformer that tries to predict state spaces without knowing about the state space equations. Do you think this would make sense?

**Audience:**

Yes

**Audience Explanation:**

This work surely has interest in the community.

**Broader Impact Concerns:**

No issues

**Claims And Evidence:**

No

**Claims Explanation:**

I'm not totally sure what the claims precisely are. We see that transformers can be trained to predict state space systems, but what does this tell us? Does the network actually use the state-space specification input in a way we think it does? The network isn't able to match exact kalman filter in performance: doesn't this mean that the a transformer can't reproduce kalman filters?

See below for more comments.

**Requested Changes:**

See below.

---

### Review · Reviewer_WLHd · 2026-01-02

**Summary Of Contributions:**

The paper explores whether transformers trained via in-context learning can exhibit behavior similar to classical filtering methods for dynamical systems. It combines constructive arguments about the expressivity of transformer primitives with extensive experiments on synthetic linear and nonlinear systems, comparing transformer predictions to Kalman, extended Kalman, and particle filters.

**Additional Comments:**

My comments are based on limited domain expertise and focus mainly on clarity and scope rather than technical correctness. Clarifying which claims are theoretical versus empirical would help non-specialist readers assess the contribution.

**Audience:**

Yes

**Audience Explanation:**

The topic is relevant to researchers working on in-context learning, algorithmic behavior of transformers, and connections between neural networks and classical estimation methods.

**Claims And Evidence:**

No

**Claims Explanation:**

The submission presents substantial empirical results and detailed theoretical constructions. However, as a non-expert in filtering and control, I found it difficult to clearly assess whether the evidence fully supports the claims. In particular, the theoretical components appear to focus on representational capacity rather than guarantees about learning or generalization, and it is not always clear how these relate to the empirical findings. The presentation is technically dense, which makes it challenging to confidently judge the strength of the evidence beyond the reported experiments.

**Requested Changes:**

- More clearly distinguish between empirical observations and theoretical claims.
- Improve accessibility by summarizing the main contributions and limitations at a higher level.
- Clarify the intended scope of the conclusions, particularly what is demonstrated empirically versus what remains conjectural.

---

### Decision · Action_Editor_QSp4 · 2026-02-25

**Recommendation:** Accept as is

**Additional Comments:**

The reviewers’ concerns have been substantially addressed in the latest revision, and the paper is technically sound and of interest to the community.

**Audience:**

Yes

**Audience Explanation:**

The paper will be of interest to a meaningful subset of the TMLR audience. The work provides a concrete bridge between modern transformer ICL behavior and state estimation in state-space models, with a careful experimental study and an explicit discussion of scope and limitations that helps accessibility.

**Claims And Evidence:**

Yes

**Claims Explanation:**

The submission combines (i) a constructive representability argument showing how Kalman-style updates can be expressed using transformer-native primitives and (ii) extensive controlled experiments on linear and nonlinear systems comparing against KF/EKF/PF, including ablations where withholding system information degrades performance. The revised manuscript also makes the scope explicit: the theory concerns representability rather than learning/generalization guarantees, and the empirical conclusions are framed accordingly. Overall, the evidence is accurate, coherent, and sufficient to support the stated claims.